# Stabilization of supramolecular membrane protein–lipid bilayer assemblies through immobilization in a crystalline exoskeleton

Fabian C. Herbert [1,5], Sameera S. Abeyrathna[1,5], Nisansala S. Abeyrathna[1,5], Yalini H. Wijesundara[1], Olivia R. Brohlin[1], Francesco Carraro[2], Heinz Amenitsch[3], Paolo Falcaro[2], Michael A. Luzuriaga[1], Alejandra Durand-Silva [1], Shashini D. Diwakara[1], Ronald A. Smaldone[1], Gabriele Meloni [1✉] & Jeremiah J. Gassensmith [1,4✉]

Artificial native-like lipid bilayer systems constructed from phospholipids assembling into unilamellar liposomes allow the reconstitution of detergent-solubilized transmembrane proteins into supramolecular lipid-protein assemblies called proteoliposomes, which mimic cellular membranes. Stabilization of these complexes remains challenging because of their chemical composition, the hydrophobicity and structural instability of membrane proteins, and the lability of interactions between protein, detergent, and lipids within micelles and lipid bilayers. In this work we demonstrate that metastable lipid, protein-detergent, and protein-lipid supramolecular complexes can be successfully generated and immobilized within zeolitic-imidazole framework (ZIF) to enhance their stability against chemical and physical stressors. Upon immobilization in ZIF bio-composites, blank liposomes, and model transmembrane metal transporters in detergent micelles or embedded in proteoliposomes resist elevated temperatures, exposure to chemical denaturants, aging, and mechanical stresses. Extensive morphological and functional characterization of the assemblies upon exfoliation reveal that all these complexes encapsulated within the framework maintain their native morphology, structure, and activity, which is otherwise lost rapidly without immobilization.

---

[1] Department of Chemistry and Biochemistry, The University of Texas at Dallas, Richardson, TX, USA. [2] Institute of Physical and Theoretical Chemistry, Graz University of Technology, Graz, Austria. [3] Institute of Inorganic Chemistry, Graz University of Technology, Graz, Austria. [4] Department of Bioengineering, The University of Texas at Dallas, Richardson, TX, USA. [5]These authors contributed equally: Fabian C. Herbert, Sameera S. Abeyrathna, Nisansala S. Abeyrathna. ✉email: gabriele.meloni@utdallas.edu; gassensmith@utdallas.edu

All living organisms, from prokaryotes to higher eukaryotes, rely on transmembrane protein systems for a variety of functions including signal transduction, substrate transport, and intramembrane enzymatic catalysis[1]. A significant fraction of polytopic transmembrane proteins act as transporters and are critical for the translocation of large and/or charged substrates in and out of the cell, and understanding their function is imperative to understanding the etiology of many human diseases[2–4]. Purification of transmembrane proteins and subsequent reconstitution in artificial lipid bilayers—called proteoliposomes—generates metastable systems that are utilized for both structural analysis and functional investigations in which substrate transport can be studied on transporters embedded into their native–like environments[5,6]. Unfortunately, owing to the patchwork of hydrophobic and hydrophilic surfaces of transmembrane proteins and the dynamic non-covalent nature of liposomes, these assemblies are intrinsically unstable and susceptible to denaturation, precipitation, and loss of function when left at room temperature for even a few hours. Despite years of effort toward stabilizing membrane-bound proteins, only a handful of approaches exist, and discovery of methods that protect them within their native bilayer environment remains an open challenge. Immobilization of biomacromolecular motion by chaperone-like confinement in polymers has emerged as a way to stabilize soluble proteins, but the accessibility of addressable functional groups of transmembrane proteins is reduced from confinement in a lipid matrix and the poor stability of membrane-bound proteins further complicates bioconjugation[7]. Recently, polymeric excipients designed to match the natural distribution of polar residues on proteins[7] or stabilization in amphipathic copolymer-based membrane nanodiscs[8] have advanced the state-of-the-art; however, these coatings are neither removable nor scalable to protect a functional catalytic proteoliposome system. Specifically, a sheddable coating that avoids

excipients or chemical functionalization of either protein or lipid would help advance work on these systems tremendously. To that end, we turned to metal–organic frameworks (MOFs), which are a class of crystalline porous coordination polymers constructed from the interlinking between metal centers and monomeric organic ligands[9–11]. Through thoughtful selection of the organic ligand and metal center, MOFs can be modulated sophisticatedly toward a wide array of applications[12] such as gas separation/sensing/storage[13–15], catalysis[16–18], and protein stabilization[19–21]. In recent years, several MOFs displaying zeolitic topologies have been reported[22], the most ubiquitous being zeolitic-imidazole framework-8 (ZIF-8) and associated structural analogues, all of which are coordination polymers consisting of $Zn^{2+}$ and 2-methyl imidazole (MIM)[23,24]. ZIF-8 and its associated (pseudo) polymorphs can form thermodynamically stable crystalline shells by nucleating on biomacromolecules[25] and these systems can withstand high temperatures and pH[26] yet are kinetically labile in the presence of metal binding chelates[27–29]. ZIF-8 in particular is well known for its ability to nucleate on the surface of colloidal and dissolved biomacromolecules forming a crystalline matrix shell[30,31]. We suspected that colloidal liposomes, proteoliposomes, and detergent-solubilized transmembrane proteins would nucleate the growth of ZIFs over their surface, and this protective shell would inhibit both protein denaturation and liposome fusion and/or degradation. Further, the kinetic lability of Zn-MIM bond in the presence of high-affinity metal chelators would allow us to recover the encapsulated systems without significant loss of protein function or sample homogeneity. We selected two α-helical polytopic (eight transmembrane α-helices) iron and copper transporters—called IroT/MavN and CopA respectively—as representative examples of metal transport systems and virulence factors in bacteria that cause fatal human diseases[32,33]. Transition metal transporter proteins are the subject of ongoing research in a number of laboratories[34–36], and their stabilization serve not only as a proof-of-concept but also in aiding and abetting research into these systems.

In this work, we demonstrate a method for the thermal stabilization of blank 200 nm liposomes, purified transmembrane proteins, and 200 nm transmembrane protein-liposome supramolecular complexes (proteoliposomes) against chemical and thermal stressors through biomimetic nucleation of a pseudopolymorph of ZIF-8 called ZIF-L. We found that encapsulation of the proteoliposome complex generates thermodynamically stable bio-composites that can withstand exposure to high temperatures, aging, and common protein denaturants (Fig. 1). Further, the ZIF-L coatings can be removed to afford pristine proteoliposomes, liposomes, and transmembrane protein–micelle complexes of similar composition, morphology, structure, and catalytic activity to their native counterparts. Finally, our work demonstrates the generalizability and potential ZIF scaffolds have for stabilizing highly sensitive and metastable supramolecular systems.

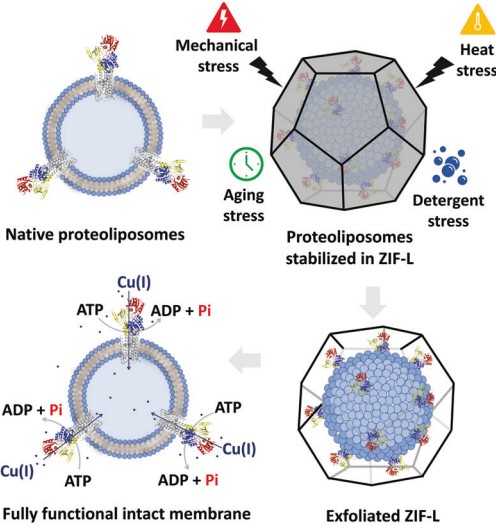

**Fig. 1 Proteoliposome stabilization through biomolecular nucleation in ZIF-L.** Proteoliposome with CopA embedded cartoon (Top left panel). Proteoliposome encapsulated in ZIF-L cartoon (Top right panel). Removal of ZIF coating from the proteoliposome (Bottom right corner). Recovered, fully functional proteoliposomes cartoon (Bottom left corner). The scheme portrays the process by which the proteoliposome after encapsulation can resist different forms of stress. An exfoliating step is then used to remove the shell and free the proteoliposomes when ready for analysis. Even after intense stressing, catalytic activity of transmembrane copper transporters remains very high.

## Results and discussion

**Liposome stabilization.** We progressed systematically to demonstrate that each component—the as-prepared liposome, the detergent purified protein, and the proteoliposome system—could be encapsulated and protected by biomolecular nucleation. We prepared 200 nm liposomes by freeze-fracture and extrusion using mixtures of L-α-phosphatidylcholine and *E. coli* total polar lipid extracts, that allow the formation of native-like unilamellar lipid bilayers. While this composition was selected because it provides the best stabilization for our selected transmembrane proteins, the anionic nature of the formulated lipids is useful in promoting ZIF nucleation and growth, as we and others have

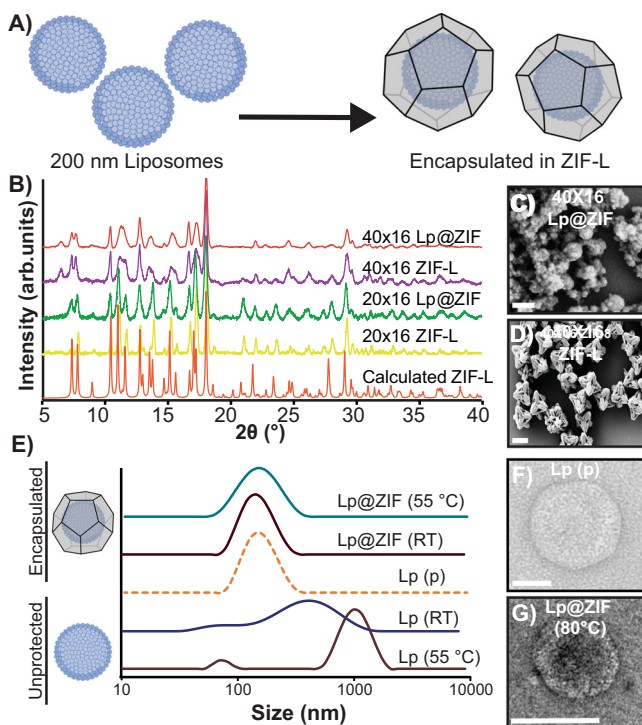

**Fig. 2 Characterization of artificial lipid bilayers embedded in ZIF.**
**A** Biomolecular nucleation of liposomes in ZIF. **B** PXRD spectra of ZIF liposome complexes (Lp@ZIF) and ZIF controls. $40 \times 16$ Lp@ZIF (red), $40 \times 16$ ZIF-L (purple), $20 \times 16$ Lp@ZIF (green), $20 \times 16$ ZIF-L (yellow), and calculated ZIF-L (orange). SEM micrograph of (**C**) $40 \times 16$ Lp@ZIF (Scale bar = 200 nm) and (**D**) $40 \times 16$ pristine ZIF (Scale bar = 2 μm). **E** DLS profiles obtained for immobilized liposomes exposed at high temperatures (after exfoliation) compared to native and nonencapsulated liposomes. Lp@ZIF 55 °C (green), Lp@ZIF RT (brown), Lp pristine (orange-dashed line), Lp RT (blue), and Lp 55 °C (dark-pink). **F** TEM micrograph of pristine liposome (Scale bar = 50 nm) and (**G**) $40 \times 16$ Lp@ZIF following exfoliation after exposure to 80 °C for 5 min (Scale bar = 100 nm).

fluorescence we determined encapsulation efficiency, where we found that both conditions afforded high liposome loading (90%). Fluorescence imaging, powder X-ray diffraction (PXRD), and scanning electron microscopy (SEM) analysis of the Lp@ZIF formulation as well as controls were carried out (Fig. 2B–D and Supplementary Figs. 1, 2). Intriguingly, XRD patterns obtained for both conditions, including controls, are consistent with the formation of ZIF-L, a crystalline phase with a leaf-like morphology that explains the highly faceted nanoparticles observed in the micrographs in Fig. 2C, D[39]. Nitrogen isotherms of both liposome loaded composites and controls reveal low porosity (Supplementary Fig. 2C and Supplementary Table 1), which is consistent with ZIF-L[40]. Finally, thermogravimetric analysis (TGA) of $20 \times 16$ Lp@ZIF and $40 \times 16$ Lp@ZIF both display a mass loss (~18%) at 200 °C, which has been attributed to solvent loss in prior work with ZIF-L (Supplementary Fig. 2D)[41].

The protective capacity of the ZIF-L coating toward the liposomes was evaluated on samples suspended in water and stressed at RT for 48 h and 55 °C for 15 min—this latter temperature being above the phase transition temperature of the liposomes. After stressing, the ZIF-L shell was dissolved in ethylenediaminetetraacetic acid-(EDTA; 50 mM)—a process we refer to as exfoliation (Fig. 1)—and these solutions were analyzed by dynamic-light scattering (DLS) to assess liposomal size distribution. As controls, we stressed nonencapsulated liposomes following the same experimental conditions. As expected, the unprotected liposomes showed a significant increase in size and polydispersity (average diameter: 604.8 nm; PDI: 0.449; Supplementary Table 2) resulting from membrane fusion and liposome aggregation (Fig. 2E, lower two traces) under both conditions. Conversely, we were happy to find that the monodisperse nature of freshly extruded blank liposomes (average diameter: 141.2 nm; PDI: 0.138) was retained for the ZIF-L coated composites (average diameter: 123.2 nm; PDI: 0.179; Fig. 2E top two traces). Preservation of the original liposomal morphology in ZIF-L immobilized samples was further confirmed by transmission-electron microscopy (TEM, Fig. 2F, G). While the stressed ZIF-L liposomes showed liposomes that were indistinguishable from freshly extruded pristine samples, unprotected liposomes showed altered morphology as a consequence of extensive membrane rupture and fusion (Supplementary Fig. 3). In light of the observed stabilization, Lp@ZIF composites were thermally stressed at 80 °C and subsequent analysis by DLS and TEM imaging revealed that the original morphology and size distribution was largely retained (average diameter: 119.3 nm; PDI: 0.231; Supplementary Fig. 4). To test the ability of ZIF encapsulation in protecting from aging, Lp@ZIF was dried and the obtained hydrated suspension left on the bench top for 20 days at room temperature. Following exfoliation, TEM characterization revealed that the liposomes retained their original size distribution and morphology (Supplementary Figs. 3, 4). In contrast, the nonencapsulated liposomes kept in solution fused and/or aggregated within 2 days (Supplementary Fig. 3F–I). Interestingly, both formulations discussed above ($20 \times 16$ and $40 \times 16$) provided outstanding protection against thermal stress and aging following liposome immobilization in ZIF-L composites (Fig. 2, and Supplementary Figs. 3, 4).

Finally, we and others have previously reported that different polymorphs of ZIF can be produced by increasing the concentration of ligand and metal[22,30]. This is potentially useful as different polymorphs can impart greater or less protection and the kinetics of dissolution can be very different, which may be useful in drug delivery applications. We were able to find conditions that resulted in the growth of the ZIF-8 sod (sodalite) topology (Supplementary Fig. 5). Though, qualitatively, we noticed it was more time consuming to exfoliate the ZIF-8

demonstrated[37,38]. Prior work with ZIF-8 synthesis has been done in pure water; however, the internal composition of the liposome lumen requires ideal buffering conditions to guarantee protein activity and stability when the proteoliposomes are generated and avoid osmotic bursting. Instead, we used a solution of 100 mM NaCl, 1 mM TCEP, and 20 mM MOPS buffered to a pH of 7.0 (M-buffer) as a solvent and systematically varied the concentrations of Zn salt and MIM until liposomes (Lp) were quantitatively captured within ZIF-L shells forming Lp@ZIF (Fig. 2A).

Our investigation focused primarily on two synthetic conditions that produced crystal encapsulated liposomes—M-buffer and 20 mM of zinc acetate dihydrate with 320 mM MIM, which is a MIM concentration 16-fold higher than the metal concentration ($20 \times 16$) and 40 mM of $Zn(OAc)_2$ with 640 mM MIM ($40 \times 16$). Precursor solutions of zinc acetate dihydrate, MIM, and 200 nm liposomes were mixed to immediately yield a white flocculate. Crystals were allowed to age at RT for 18 h, then the artificial lipid bilayer-embedded ZIF crystals were harvested by centrifugation, washed, and allowed to dry at RT for 12 h. Encapsulation efficiency was determined by fluorescence. Cyanine-5 (Cy5, $\lambda_{Ex} = 651$ nm and $\lambda_{Em} = 670$ nm) was entrapped within the liposomal lumen during liposome extrusion. After washing, the fluorescent liposomes were encapsulated using both the $20 \times 16$ and $40 \times 16$ conditions (Supplementary Fig. 1). The supernatants from the reactions were collected and from the residual

polymorph as compared to ZIF-L. That said, we found we could recover the liposomes intact (Supplementary Fig. 6). Further study is underway to fully evaluate the relative levels of protection afforded by the different polymorphs of ZIF and the kinetics of their exfoliation; however, the ZIF-L composites provided outstanding protection and were used throughout the work.

**Mechanism of ZIF growth.** In reactions with ZIF and its associated polymorphs, we and others have observed that the initial interaction occurs between the biomolecule surface and the zinc ions and this interaction has been proposed as an important indicator for a biomimetic crystallization process on the surface of biomacromolecules[26,30]. We were able to confirm qualitatively that zinc can bind to the negative surface of liposomes, which have a $-16.9$ mV $\zeta$ potential (Supplementary Fig. 7A, B) from the presence of anionic phospholipids, by observing zinc-lipid complexes by inductive coupled plasma mass spectrometry. When the lipids were doped with 1,2-dioleoyl-3-trimethylammonium propane (DOTAP), a cationic lipid, we observed less or even no $Zn^{2+}$ interaction with the liposome (Supplementary Fig. 7C, D). Cationic surface charge has been shown to reduce the encapsulation yield or even prevent crystal growth, although there are strategies to overcome this, and we are actively studying this. Kinetics of nucleation, particle growth[38], crystallization, and the morphology of the particles were investigated in situ via synchrotron-based small-angle and wide-angle X-ray scattering (SAXS/WAXS) techniques[42,43]. We investigated four different samples ($20 \times 16$ Lp@ZIF, $20 \times 16$ ZIF-L, $40 \times 16$ Lp@ZIF, and $40 \times 16$ ZIF-L) using a stopped flow device to initiate the rapid mixing of the reagents (mixing time <100 ms). See the SAXS section in methods for full experimental details, and Supplementary Figs. 8–12 and Supplementary Tables 5, 6 for experimental setup and results. The injection of the aqueous precursors solutions ($Zn^{2+}$, MIM, liposomes) into a micromixer triggered the SAXS acquisition system data collection (time resolution: 100 ms). Data for the $20 \times 16$ conditions are summarized in Fig. 3— see the Supplementary Information for $40 \times 16$ results. To investigate the kinetics of nucleation and growth, we monitored the time evolution of SAXS patterns (Fig. 3A, B) and of the invariant $\tilde{Q}$ (Fig. 3C, see Methods for details)[41]. $\tilde{Q}$ is related to the Porod invariant of the scattering curve and is sensitive to changes in particle volume fraction and electron density contrast. An increase of $\tilde{Q}$ over time indicates the formation of particles/ agglomerates within the investigated volume of the sample. A plateau in the time series of $\tilde{Q}$ values indicates stationary conditions. The time evolution of $\tilde{Q}$ is reported in Fig. 3C and in the "Time-resolved Small Angle X-Ray Scattering (SAXS)" section of Methods. The increase of $\tilde{Q}$ related to the particle growth is observed 0.8 and 0.6 s after the mixing of precursors for samples $20 \times 16$ Lp@ZIF and $40 \times 16$ Lp@ZIF, respectively. In the control samples, the particle growth is observed at 4 s ($20 \times 16$ ZIF-L) and 2.6 s ($40 \times 16$ ZIF-L) after mixing the precursors. The plateau of $\tilde{Q}$ related is reached 25 and 5 s after the mixing of precursors for samples $20 \times 16$ Lp@ZIF and $40 \times 16$ Lp@ZIF, respectively. In the control samples, the plateau is reached 40 s ($20 \times 16$ ZIF) and 25 s ($40 \times 16$ ZIF) after mixing the MOF precursors.

The crystallization kinetics were monitored by following the integrated intensity of the (200) ZIF-L diffraction peak (5.25 nm$^{-1}$; Fig. 3D, E). In the presence of liposome, the added mixture of Zn and MIM initially produced an amorphous phase for both the $20 \times 16$ and $40 \times 16$ conditions between 0.1–120 and 0.1–50 s respectively. The initial formation of an amorphous phase is consistent with what we have observed with viral nanoparticles and other proteins[30,42,44]. Importantly, compared to the control samples, we observed faster crystallization for Lp@ZIF particles

(e.g., $20 \times 16$ Lp@ZIF crystallization is 15 times faster than the pure ZIF-L particles; See below). These data demonstrate that the presence of liposomes templates a faster nucleation, growth, and crystallization of ZIF-L when compared with the control conditions.

By fitting the SAXS patterns 120 s after the mixing of the reagents (Fig. 3F), we observed that the presence of the liposome induced the formation of plate-like particles with a thickness of 30–50 nm. Conversely, in the absence of liposomes, ZIF particles with an average size larger than 100 nm and no sharp size distribution were observed. Thus, a role of the biomacromolecules in the final crystal morphology was determined[26]. From these data, we conclude that the liposomes act as seeding agents for the MOF growth, altering their growth kinetics as well as the ultimate morphology of the crystalline particles that are produced and that liposome MOF biocomposites are formed via a biomimetic mineralization process[43,45].

**Stabilization of purified transmembrane proteins.** We selected two different transmembrane proteins, both of which are poorly stable at room temperature, to demonstrate the broad utility of our approach. IroT/MavN is a transmembrane protein found in *Legionella pneumophila* (*L. pneumophila*), a thin, flagellated, gram-negative bacteria responsible for Legionnaires' disease[46]. IroT mediates iron sequestration as an essential micronutrient from host cell, allowing for *L. pneumophila* to replicate in a host-derived vacuole within the infected macrophages[47]. IroT topology is characterized by eight transmembrane (TM) helices and a long C-terminal domain[48,49]. The structure and substrate translocation modality in IroT are active areas of research, but much has been gleaned from reconstituting IroT in artificial lipid bilayer systems and performing real-time transport assays[49]. IroT was shown to act as a $Fe^{2+}/H^+$ antiporter that allows $Fe^{2+}$ acquisition into the vacuole from the host cell for pathogen survival[48]. The second protein selected is a copper $P_{1B}$-type ATPase from *E. coli* (CopA), a TM primary-active pump, and part of the P-type ATPase superfamily, that utilize energy generated by ATP hydrolysis to drive $Cu^+$ transport across biological membranes against electrochemical gradients[50,51]. These catalytically driven pumps constitute an essential system to drive the selective translocation and export of $Cu^+$ ions, thereby controlling the intracellular $Cu^+$ levels[52,53]. Their activity tightly balances the biogenesis and integrity of copper centers in vital enzymes to nontoxic intracellular copper levels. The CopA structure is characterized by the existence of an 8 TM helices membrane domain (M-domain) connected to large cytosolic domains (N-, P-, and A-domains) responsible for ATP hydrolysis, phosphorylation and energy transduction, allowing $Cu^+$ translocation across the membrane[35,50,52,54]. As a result of their critical involvement in essential iron and copper metabolism, both IroT and CopA homologs have been identified as key virulence factors in bacterial pathogens[37,55].

TM proteins, including IroT and CopA, are commonly extracted from membranes and purified as detergent micellar complexes for solubilization in aqueous environments. The detergent molecules surround the hydrophobic regions of the protein in the micelles, which helps avoid aggregation, precipitation, and unfolding in water. Though they are more stable, these proteinaceous assemblies still require unique environmental conditions to remain fully active—e.g., long-term storage at $-80$ °C, constant refrigeration for analysis, etc. Since this strategy is employed in the typical workflow for incorporating TM proteins in liposomes[39,47], and naked TM proteins are extremely prone to denaturation, we suspected simply nucleating the ZIF over the detergent–protein supramolecular complex would

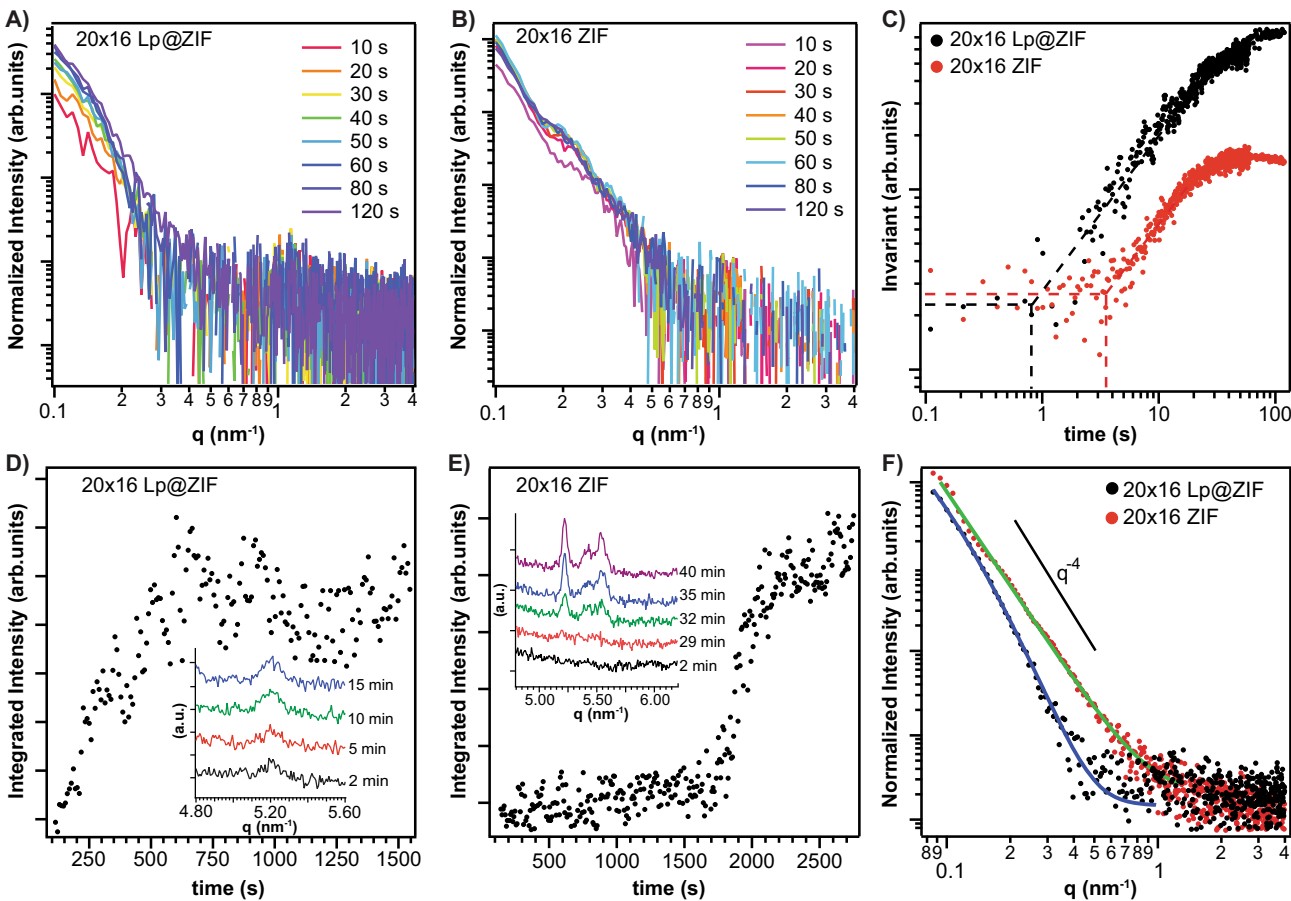

**Fig. 3 X-Ray Diffraction Kinetics Study of Biomimetic ZIF Formation.** Time evolution of SAXS patterns (background subtracted) from time-resolved SAXS synthesis of (**A**) 20 × 16 Lp@ZIF. 10 s (magenta), 20 s (orange), 30 s (yellow), 40 s (green), 50 s (blue), 60 s (dark-blue), 80 s (light-purple), and 120 s (dark-purple). **B** 20 × 16 ZIF measured at 10 s intervals. 10 s (pink), 20 s (magenta), 30 s (dark-orange), 40 s (light-orange), 50 s (green), 60 s (blue), 80 s (dark-blue), and 120 s (purple). **C** Time evolution of invariant $\tilde{Q}$ calculated from 0.1 to 0.6 nm$^{-1}$ of the time-resolved SAXS patterns in (**A**, **B**) and corresponding to the synthesis of 20 × 16 Lp@ZIF (black dots) and 20 × 16 ZIF-L (red dots). The dashed lines are plotted to highlight the starting time of the $\tilde{Q}$ increase. **D** Time evolution of the integrated intensity of (200) diffraction peak of ZIF-L (5.25 nm$^{-1}$) calculated from time-resolved SAXS synthesis 20 × 16 Lp@ZIF and (**E**) 20 × 16 ZIF-L. In the insets, selected diffraction patterns highlighting the time-evolution of the (200) diffraction peak of ZIF-L (5.25 nm$^{-1}$) are reported. Color scheme for (**D**): 2 min (black), 5 min (red), 10 min (green), and 15 min (blue), and E: 2 min (black), 29 min (red), 32 min (green), and 35 min (blue), and 40 min (purple). Time zero is referred to the end of the precursors mixing. **F** SAXS patterns (background subtracted and averaged) and fitted data 120 s after mixing the precursors of 20 × 16 Lp@ZIF (black dots) and 20 × 16 ZIF-L (red dots). The theoretical Porod power law ($I(q) \propto q^{-4}$) is plotted for comparison.

improve the likelihood of retaining protein function in high yields following exfoliation. We thus solubilized and purified IroT in Cymal-7 (7-Cyclohexyl-1-Heptyl-β-D-Maltoside) micelles and CopA in micelles prepared with DDM (n-Dodecyl-β-D-Maltopyranoside) and applied our synthetic strategy, developed above, to produce ZIF-L composites (Fig. 4A). Crystals were isolated by centrifugation, washed, and allowed to dry at RT for 12 h. As-obtained crystals were characterized by SEM and showed a star-shaped morphology (Fig. 3B–E and Supplementary Fig. 13A, B) and PXRD of both protein–micelle composites and controls again showed the phases to be ZIF-L (Fig. 4F). TGA analysis of 40 × 16 IroT@ZIF and 40 × 16 control revealed high thermal stability as shown in Supplementary Fig. 13, where the 40 × 16 IroT@ZIF is characterized by an ~20% mass loss at 200 °C, attributed to loss of solvent (Supplementary Fig. 13C)[41]. Further, nitrogen isotherms of 40 × 16 IroT@ZIF reveal no measurable porosity, whereas 40 × 16 ZIF-L shows a BET surface of 385 m$^2$g$^{-1}$ (Supplementary Table 1).

Quantification of encapsulation efficiency was determined by SDS-PAGE gel densitometry. Supernatants obtained during the washing of ZIF-L bio-composites and exfoliated protein–detergent complexes were run in tandem with either IroT or CopA pristine

standards of varying concentrations. We found the encapsulation efficiency to be quantitative—no residual protein was found in the supernatant after the encapsulation procedure. Indeed, after encapsulation, isolation of the final protein@ZIF product, and subsequent exfoliation, resulted in almost 75% recovery, regardless of the protein or metal-to ligand ratio used (Supplementary Fig. 14A–C). To determine the integrity of IroT or CopA detergent–protein micelles after biomolecular nucleation, two properties were analyzed to benchmark the protective effect of immobilization: monodispersity analysis by size-exclusion chromatography (SEC) for IroT; and catalytic metal transport activity assessed by metal-stimulated ATP-hydrolysis, for CopA. SEC is a good proxy for testing the stability of the generated IroT–detergent complexes as the absence of aggregation is an indicator of the integrity and stability of the protein–detergent assembly. On the other hand, for purified Cu$^+$ P-type ATPases, we can measure the rate of ATP hydrolysis in the presence of selected metal substrate to show persistence of structure and function, as ATP hydrolysis and Cu$^+$ transport in CopA are tightly coupled. To verify CopA functionality in detergent micelles (or upon incorporation in proteoliposomes) the Cu$^+$-dependent stimulation of ATPase activity was determined by photometric quantification

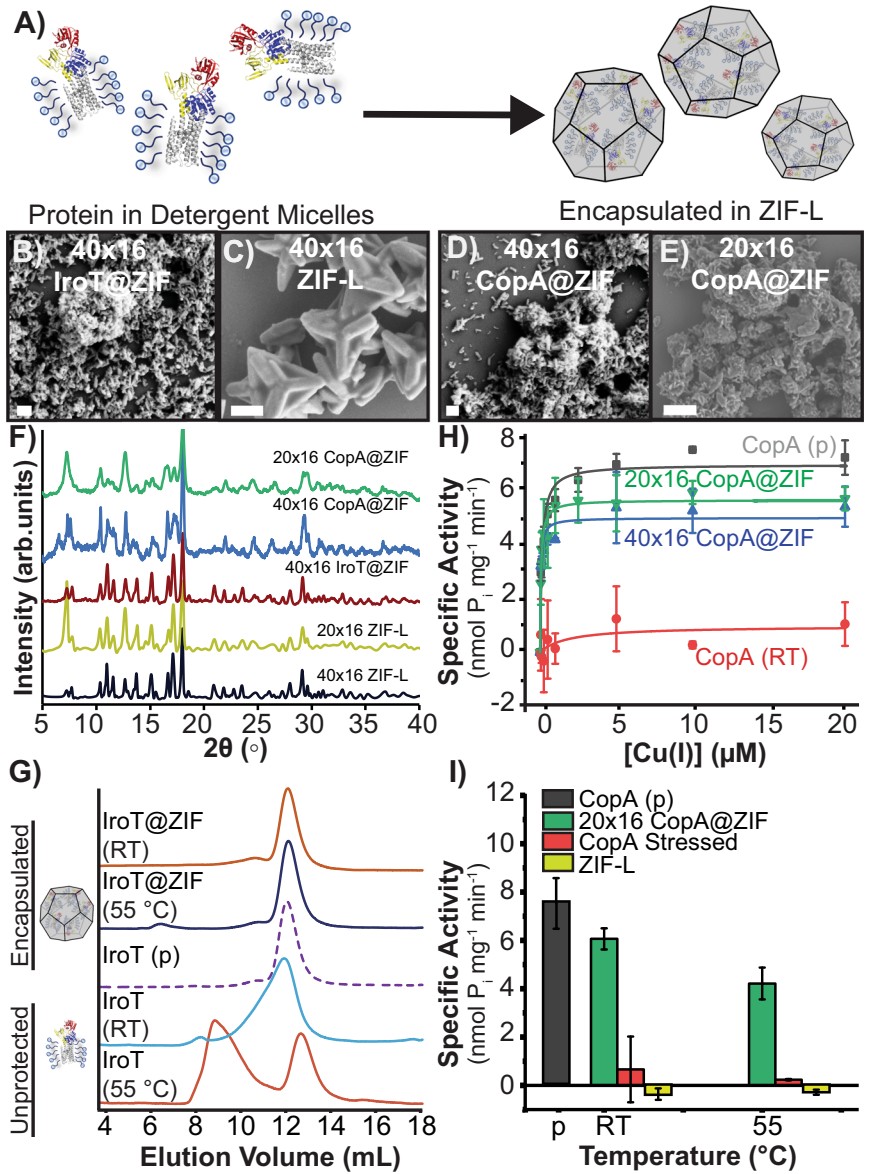

**Fig. 4 IroT@ZIF and CopA@ZIF characterization. A** Biomolecular nucleation of detergent stabilized CopA and IroT in ZIF. SEM micrographs of (**B**) 40 × 16 IroT@ZIF (Scale bar = 200 nm) and (**C**) 40 × 16 ZIF-L (Scale bar = 1 μm). SEM micrographs of (**D**) 40 × 16 CopA@ZIF (Scale bar = 200 nm) and (**E**) 20 × 16 CopA@ZIF (Scale bar = 1 μm). **F** PXRD of 40 × 16 IroT@ZIF (brown), 40 × 16 CopA@ZIF (blue), 20 × 16 CopA@ZIF (green), 20 × 16 ZIF-L (yellow), and 40 × 16 ZIF-L (black). Y-axis expressed as arbitrary units (arb. units). **G** SEC traces of heated IroT@ZIF bio-composites against native and nonencapsulated IroT. IroT@ZIF RT (orange), Lp@ZIF 55 °C (blue), IroT pristine (purple-dashed line), IroT RT (blue), and IroT 55 °C (dark-orange). **H** ATPase activity analysis of CopA pristine (gray) against stressed and exfoliated 40 × 16 CopA@ZIF (blue), 20 × 16 CopA@ZIF (green), with unencapsulated control (red line) (Error bars = standard deviation (n = 3)). **I** Comparison of maximal specific activity in CopA samples and controls (Error bars = standard deviation (n = 3)). CopA pristine (gray), 20 × 16 CopA@ZIF (green), unencapsulated CopA (red), and ZIF-L pristine (yellow).

of released inorganic phosphate ($P_i$) generated by catalytic ATP-hydrolysis using Malachite green. Upon stressing, immobilized samples were exfoliated and immediately characterized by either SEC or metal-dependent ATP-hydrolysis assays. The nonencapsulated IroT samples showed increased polydispersity when incubated at RT, as evidenced by development of an asymmetric elution peak shoulder in SEC and almost complete aggregation after exposure to 55 and 80 °C for a few minutes (Fig. 4G and Supplementary Fig. 15A). This is to be expected, as these proteins are extremely prone to aggregation at even low temperature. In contrast, for the encapsulated samples, exposure to RT for 48 h and 55 °C for 15 min had little impact on the monodispersity of IroT. SEC analysis of the exfoliated IroT samples show a single elution peak (elution volume = 12.0 mL), closely corresponding to

non-stressed and refrigerated IroT controls (12.0 mL) as shown in Fig. 4G. Incredibly, even exposure to 80 °C (5 min) produced minimal aggregation (Supplementary Fig. 15A).

Resilience to thermal stress was subsequently investigated on the CopA-DDM micelle complexes by analyzing the catalytic substrate-dependent ATP hydrolytic activity after exposing the various samples suspended in water to a range of denaturing temperatures. $Cu^+$-dependent stimulation of ATPase rates for non-stressed CopA, in the presence of $Mg^{2+}$/ATP, revealed characteristic catalytic hyperbolic Michaelis–Menten-type dependency as a function of $Cu^+$, confirming that the purification in detergent micelles maintains CopA in a functional form ($K_{M, Cu(I)} = 0.12 \pm 0.02$ μM; $V_{max}$ of $7.1 \pm 0.2$ nmol (mg min)$^{-1}$). However, upon thermal stress at RT, 55 and 80 °C, the CopA-DDM

catalytic activity was completely abolished, with <10% residual activity at RT. In contrast to unprotected protein, stressed samples of CopA encapsulated in ZIF retained the characteristic Michaelis–Menten dependency of their ATPase activity after thermal stress and exfoliation (Fig. 4H). Analysis of CopA@ZIF bio-composite ATPase activities at saturating $Cu^+$ concentrations revealed that CopA retained >80% (6.3 nmol $P_i$ $mg^{-1}$ $min^{-1}$) of its maximal ATPase activity upon stress at RT (Fig. 4I), >60% at 55 °C (4.7 nmol $P_i$ $mg^{-1}$ $min^{-1}$; Fig. 4I) and at >42% at 80 °C (3.1 nmol $P_i$ $mg^{-1}$ $min^{-1}$; Supplementary Fig. 14D). Accordingly, analysis of the $K_{M, Cu(I)}$ values at RT indicated that upon stress unaltered catalytic parameters are preserved by ZIF encapsulation (Supplementary Fig. 14E).

It is noteworthy that formulation conditions are an important aspect of stability. CopA was encapsulated under both metal-to-ligand ratios discussed above ($20 \times 16$ and $40 \times 16$) and, interestingly, IroT-cymal-7 micelles showed better stabilization with the $40 \times 16$ formulation, while the $20 \times 16$ formulation was most effective at enhancing the thermal stability of CopA-DDM micelles. Thus, formulation optimization is an important parameter to be screened for optimal bio-composite protection depending on the protein topology and protein–detergent micelle structure (Supplementary Fig. 14D–F).

Encapsulation of proteinaceous materials has been widely used to increase stability of moieties against chemical stressors, such as organic solvents and chaotropic agents[33,34]. Motivated by such reports, CopA@ZIF and IroT@ZIF were chemically stressed using SDS, a commonly used protein denaturant. Briefly, the samples were incubated for 30 min in a solution consisting of 0.1% SDS. Crystals were harvested by centrifugation, washed 5× with ultrapure water, exfoliated, and immediately characterized by SEC analysis (IroT) or $Cu^+$-dependent ATP-hydrolysis assays (CopA). As shown in Supplementary Fig. 15, immobilization in ZIF affords retention of monodispersity and activity for both encapsulated TM proteins, while nonencapsulated control samples are fully denatured and inactive in the presence of 0.1% SDS. These results also suggest that a population of proteins are at least partly exposed to the MOF surface, accounting for the modest (~15%) loss of functionality[36].

**Proteoliposome stabilization.** Our analysis demonstrates that immobilization in new ZIF composites allow for stabilization of both pristine liposomal vesicles as well as protein–detergent micellar complexes, providing a 3D scaffold that can morph around complex and chemically diverse biomolecular assemblies providing protection against stressors. In light of the versatility of the approach, we sought to determine if our ZIF encapsulation strategy could protect even more complex and metastable supramolecular entities such as proteoliposomes. Purified IroT and CopA were reconstituted in unilamellar liposomes via freeze–thaw and extrusion through 200 nm filters, followed by liposome destabilization by detergent addition and subsequent detergent removal by Biobeads resin. Protein incorporation was subsequently quantified by SDS-PAGE following removal of excess detergent-solubilized protein from the proteoliposomes by ultracentrifugation with subsequent protein quantification of the soluble and proteoliposome fractions conducted by gel densitometry. We subsequently biomimetically mineralized proteoliposomes with IroT or CopA embedded in the lipid bilayer in ZIF (Fig. 5A–C and Supplementary Figs. 16, 17). In a typical experiment, 6.25 mg $mL^{-1}$ of proteoliposomes (protein concentration is 0.25 mg $mL^{-1}$; TEM of typical samples shown in Fig. 5B) were encapsulated in ZIF at ambient conditions mixing the proteoliposome complexes with a solution of MIM and zinc acetate dihydrate using M-buffer as solvent. Crystals were harvested by centrifugation after 18 h of aging at RT, washed with ultrapure water and allowed to dry at RT for 12 h (Fig. 5C). SDS-PAGE gel densitometry revealed protein recovery after exfoliation with no detectable traces of protein in the supernatants (Supplementary Fig. 16A). In the presence of sodium chloride, TCEP or DTT, and MOPs the crystal morphology appears as aggregates of star-like shape chunks (Supplementary Fig. 16B–H) of ZIF-L. The encapsulation and exfoliation process, which were optimized in the prior two studies, was very straight forward for proteoliposome assemblies to generate IroTPL@ZIF and CopA-PL@ZIF bio-composites. Following exfoliation, we found the proteoliposome size and shape were not altered compared to freshly extruded proteoliposomes by TEM and DLS analyses, even after heat exposures, regardless of the ZIF formulation utilized for immobilization (Fig. 5E, Supplementary Fig. 17, and Supplementary Tables 3, 4).

In addition to the size and morphology preservation, activity assays performed on the ZIF immobilized CopA proteoliposomes confirmed that the MOF shell thermally enhances these delicate systems enabling them to resist temperatures that would otherwise promote loss of function (Fig. 5F–I). Indeed, proteoliposome formulations of CopA are so thermally unstable they lose 90% of their catalytic activity in 48 h at 4 °C (Supplementary Fig. 18).

To demonstrate that the encapsulated proteoliposomes not only resist prolonged periods of no refrigeration but also enhances stability towards physical/mechanical stressors, prepared PL@ZIF samples were shipped across America in a padded envelope through the United States Postal Service. PL@ZIF bio-composites in water were placed in a standard cushioned mailer and shipped across the United States, from Dallas, Texas to Rhode Island and back again. They were then left at room temperature for 2 months following shutdown of our laboratories during the 2020 SARS-CoV-2 pandemic. By the time we had opened the package, the water had completely evaporated. Nonetheless, after exfoliation, the catalytic activity and liposome morphology were similar to the pristine counterparts (Supplementary Fig. 19) in contrast to controls, which we know degrade within a day at room temperature or 2 days under refrigerated conditions.

Lipid bilayers are the core building unit of cell membranes, which serve as the main line of action between the outside and the inside environments of the cells and organelles. Given their structural complexity, researchers have been motivated to develop simpler model systems to understand the molecular processes associated with cellular membrane dynamics and investigate protein-mediated solute translocation across lipid bilayers. Proteoliposomes are a powerful tool that mimic cellular membranes. By virtue of tuning the vesicle size and the lipid and protein composition, proteoliposomes have become instrumental to the study both prokaryotic and complex eukaryotic cell membranes, and proteins embedded into them, including TM transporter proteins. Despite their utility, proteoliposomes are delicate systems that require unique conditions to maintain their functionality that have long imposed obstacles for their handling/transport and hindering their usefulness for the better understanding of the modus operandi of TM proteins. Taking advantage of the high thermal and aqueous stabilities of ZIF-L, blank liposomes, detergent-solubilized proteins, and proteoliposome complexes no longer require constant refrigeration and repeated extrusion to maintain their intrinsic structure, monodispersity and functionality over long incubation times. Further, we show that immobilization in ZIF-L enables the as-prepared bio-composites to be exposed to chemical denaturants and temperatures above their lipid bilayer phase transition without structural and/or functional changes. Finally, we have shown that

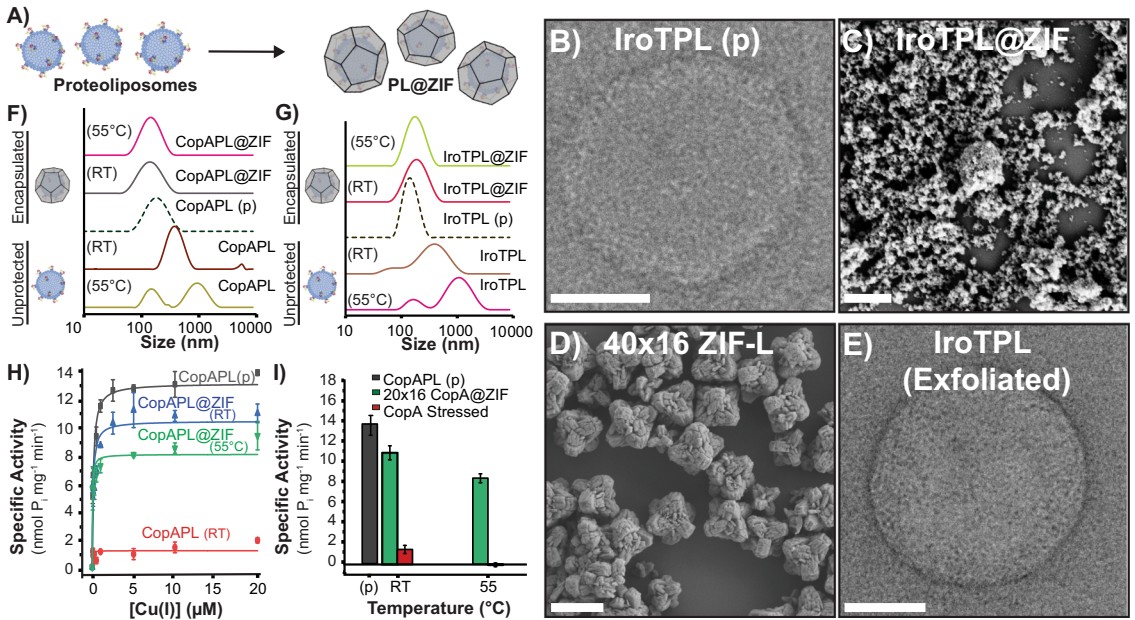

**Fig. 5 Characterization of proteoliposome@ZIF bio-composites. A** Biomolecular nucleation of CopA-PL and IroTPL in ZIF-L. **B** TEM micrograph of pristine IroT containing proteoliposomes. (Scale bar = 50 nm) **C** SEM micrograph of IroT-PL@ZIF bio-composites.(Scale bar = 1 µm). **D** SEM micrograph of ZIF-L control. (Scale bar = 1 µm). **E** TEM micrograph of a recovered fully functional IroT containing proteoliposome upon complete exfoliation. (Scale bar = 100 nm). DLS of (**F**) 20 × 16 CopAPL@ZIF bio-composites after thermal stressing followed by exfoliation compared to controls. CopAPL@ZIF 55 °C (magenta), CopAPL@ZIF RT (gray), CopAPL pristine (green-dashed line), CopAPL RT (brown), and CopAPL 55 °C (light-green). **G** 40 × 16 IroTPL@ZIF bio-composites after temperature stressing followed by exfoliation compared to controls. IroTPL@ZIF 55 °C (green), IroTPL@ZIF RT (red), IroTPL pristine (brown-dashed line), IroTPL RT (brown), and IroTPL 55 °C (magenta). **H** Specific activity of 20 × 16 CopA-PL@ZIF bio-composites stressed at RT and 55 °C (Error bars = standard deviation ($n = 3$)). CopAPL pristine (gray), 20 × 16 CopAPL@ZIF RT (blue), 20 × 16 CopAPL@ZIF 55 °C (green), and unencapsulated control (red line). **I** Comparison of maximal specific activity at saturating $Cu^+$ concentration of 20 × 16 CopA-PL@ZIF bio-composites stressed at RT and 55 °C to the nonencapsulated and stressed CopA-PL (Error bars = standard deviation ($n = 3$)). CopAPL pristine (gray), 20 × 16 CopAPL@ZIF (green), and unencapsulated control (red line).

biomolecular nucleation is an effective process to preserve supramolecular membrane protein–lipid bilayer assemblies against conditions that, without encapsulation, would easily impair their structural and functional integrity.

## Methods

**Expression and purification of MavN/IroT.** MavN/IroT protein was expressed recombinantly in *E. coli* BL21-Gold(DE3) cells using a pET-52b (+) plasmid (Genscript Inc.) encoding the codon-optimized *Legionella pneumophila* IroT sequence (strain Philadelphia 1/ATCC 33152/DSM 7513; Gene: lpg_2815)[49]. The synthetic DNA construct encoding for MavN/IroT was generated using the primers reported in Supplementary Table 7, and cloned in the pET-52b (+) SmaI/SacI restriction sites (Genscript Inc.). The plasmid was transformed in *E. coli* BL21-Gold (DE3) cells and O/N pre-cultures were inoculated at 1% (v/v) in TB media supplemented with 1% (v/v) glycerol and 50 µg mL⁻¹ ampicillin and grown at 37 °C. Cells were subsequently cooled to 18 °C after OD₆₀₀ reached a value of 2. Isopropyl thiogalactopyranoside (IPTG) was added to a final concentration of 0.30 mM to the culture to induce the protein expression. Protein expression was carried out at 18 °C for 19 h and the cells were harvested by centrifugation (20 min, 4 °C, 20,000 × g) (Thermo Scientific Sorvall LYNX 6000 centrifuge). Membranes were isolated upon harvesting the cells and resuspending them in buffer (20 mM Tris/HCl pH = 8, 150 mM NaCl, 5 mM MgCl₂, 30 µg/mL deoxyribonuclease I from bovine pancreas (Sigma-Aldrich), and 2× EDTA-free protease inhibitor cocktail tablets (Thermo Scientific)), and lysing them in an ice-chilled microfluidizer at 20,000 psi (Microfluidics M-110P) through a Z-shaped diamond chamber. Cell debris was removed by centrifugation of the lysate (20 min, 4 °C, 20,000 × g, Thermo Scientific Sorvall LYNX 6000 centrifuge). The supernatant was subjected to ultracentrifugation (1 h, 4 °C, 205,100 × g, Beckman Optima XPN80) to isolate the membrane fraction carrying IroT. The membrane pellet was suspended to a final 1 g of original cells per mL in buffer (20 mM Tris/HCl pH = 8, 500 mM NaCl, supplemented with 1% (w/v) glycerol and EDTA-free protease inhibitor cocktail).

IroT was extracted from 10 mL of membrane suspension by vigorous stirring for 1 h at 4 °C in the presence of 1% (w/v) n-Tetradecylphosphocholine detergent (Fos-Choline-14, (Anatrace)), in ice-chilled extraction buffer containing 20 mM Tris/HCl pH = 8, 500 mM NaCl, 35 mM imidazole, 5 mM β-mercaptoethanol, and EDTA-free protease inhibitor cocktail (Thermo Scientific). Unextracted proteins

and residual membrane fractions were removed by ultracentrifugation (20 min, 4 °C, 205,100 × g, Beckman Optima XPN80). The supernatant was used for the purification of IroT protein.

IroT purification was carried out by affinity chromatography using a Ni-NTA column preequilibrated in a buffer preequilibrated in Cymal-7 detergent (20 mM Tris/HCl, pH 8, 500 mM NaCl, 35 mM imidazole, 1 mM dithiothreitol (DTT), 0.05% (w/v) 7-cyclohexyl-1-heptyl-β-D-maltoside (Cymal-7). The column washing was done with 50 CV wash buffer (20 mM Tris/HCl pH = 8, 500 mM NaCl, 35 mM imidazole, 1 mM dithiothreitol (DTT), 0.05% (w/v) 7-cyclohexyl-1-heptyl-β-D-maltoside (Cymal-7) to remove any unbound impurities. The bound protein was eluted using a linear imidazole gradient (0–100%) of a mixture of the wash and elution buffer (20 mM Tris/HCl pH = 8, 500 mM NaCl, 500 mM imidazole, 0.05% (w/v) Cymal-7, 1 mM DTT). The protein containing fractions were immediately passed through a HiPrep 26/10 desalting column (GE Healthcare) on a AKTA Pure FPLC system (GE Healthcare) to exchange the buffer to a final 20 mM MOPS/NaOH pH = 7, 500 mM NaCl, 1 mM dithiothreitol (DTT), 0.05% (w/v) Cymal-7, and remove imidazole. Size-exclusion chromatography (SEC) was carried out using Superdex 200 10/300 or HiLoad Superdex 200 16/600 columns (GE Healthcare) in 20 mM MOPS/NaOH pH = 7, 500 mM NaCl, 1 mM dithiothreitol (DTT), 0.05% (w/v) Cymal-7, to remove impurities and aggregated protein, as reported earlier[2]. Protein was concentrated to ~5–10 mg/mL for incorporation in proteoliposomes using 100 kDa molecular weight cut-off (MWCO) filters (Sartorius VIVASPIN 20) by centrifugation (2100 × g, 4 °C). Purity of incorporated IroT protein was determined using SDS-PAGE (4–15% Tris-Glycine Mini-PROTEAN gels, BioRad). Absrobtion at 280 nm utilizing the extinction coefficient of ε₂₈₀ = 122,395 M⁻¹ cm⁻¹ was used to determine the protein concentration. Freshly purified protein was incorporated in proteoliposomes without freezing. SEC was carried out on IroT encapsulated in ZIF and exfoliated, to determine the monodispersity of protein encapsulated in ZIF upon stress.

**Reconstitution of MavN/IroT in proteoliposome small unilamellar vesicles (SUVs).** All the buffer solutions utilized in proteoliposome preparation were treated with Chelex resin (Biorad) to remove metal contaminants. Purified IroT was reconstituted at a protein-to-phospholipid ratio of 1:25 (w/w) as reported earlier[49]. *E. coli* polar lipids (Avanti Polar Lipids) and L-α-phosphatidylcholine (from chicken egg, Avanti Polar Lipids) in chloroform were mixed at at a 3:1 ratio (w/w) in a pear-shaped flask and a thin film was prepared by drying under a N₂ stream with rotation. The thin lipid film was further dried overnight in a vacuum

desiccator. The following day, lipids were hydrated and suspended 1 mM TCEP in $H_2O$. The lipid suspension was buffered by adding a 10× buffer stock to final 20 mM MOPS/NaOH pH = 7, 100 mM NaCl, 1 mM TCEP at a final lipid concentration of 25 mg mL$^{-1}$.

SUVs were prepared by subjecting the lipid suspension to three freeze–thaw cycles in liquid nitrogen and 11 sequential extrusions through Track-Etched polycarbonate (PC) membranes (Whatman) of decreasing pore sizes (1 μm, 400 nm, and 200 nm by utilizing a 1 mL gastight syringe system (Avanti Polar lipids)). The prepared SUVs were destabilized with detergent by adding 0.02% Cymal-7 (Anatrace) and tilting at RT for 1 h. Purified IroT (~5–10 mg mL$^{-1}$) was added at a 1:25 protein:lipid ratio (w/w) to the detergent destabilized SUVs on ice, and tilted at 4 °C for 1 h. Detergents were removed by adding Bio-Beads (SM-2; Bio-Rad) activated by washings with methanol, ethanol, and water sequentially and filtered to remove $H_2O$. Bio-Beads were added to generate a final slurry of 40 mg mL$^{-1}$. The complete detergent removal was achieved by Bio-Beads exchange after 1, 12, 14, and 16 h. The proteoliposomes were collected by ultracentrifugation at 160,000 × g at 4 °C in a Sorvall mX120+ Micro-Ultracentrifuge. The proteoliposome pellets were resuspended in buffer containing 20 mM MOPS/NaOH pH = 7, 100 mM NaCl, 1 mM TCEP (treated with Chelex) to a final lipid concentration of 25 mg mL$^{-1}$ (IroT = 1 mg mL$^{-1}$). Incorporation of IroT in proteoliposomes was determined by SDS-PAGE (4–15% Tris-Glycine Mini-PROTEAN gels, BioRad) through analysis of the supernatant and the resuspended proteoliposome pellets. Flash-frozen proteoliposomes were stored at −80 °C prior to encapsulation in ZIF.

**Expression and purification of CopA protein.** For the recombinant expression of CopA, a pET-52b (+) expression plasmid with a N-terminal Strep tag II encoding the codon-optimized CopA homolog from *E. coli* (Strain K12; UniProt accession number: Q59385) was generated by Genscript Inc[3]. The synthetic DNA construct encoding for CopA was generated using the primers reported in Supplementary Table 8 and cloned in the pET-52b (+) SmaI/SacI restriction sites (Genscript Inc.). *E. coli* CopA expression was performed by transforming the expression plasmid into *E. coli* BL21-Gold(DE3) competent cells (Agilent Technologies). The overnight preculture was inoculated in fresh TB media supplemented with 1% (v/v) glycerol and 50 mg mL$^{-1}$ ampicillin and cells were grown at 37 °C under agitation until they reached OD$_{600}$ = 2. Isopropyl thiogalactopyranoside (IPTG) was added to the cell cultures to a final concentration of 0.30 mM upon cooling to 25 °C to induce protein expression. Cell cultures were incubated at 25 °C for 18 h and cells harvested by centrifugation (20 min, 4 °C, 20,000 × g; Thermo Scientific Sorvall LYNX 6000 centrifuge).

Membranes were prepared similarly to IroT in lysis buffer (20 mM Tris/HCl pH = 8, 150 mM NaCl, 5 mM MgCl$_2$, 30 μg/mL deoxyribonuclease I from bovine pancreas (Sigma-Aldrich), and supplemented with 2× EDTA-free protease inhibitor cocktail tablets (Thermo Scientific)). Cell were lysed with an ice-cold microfluidizer at 20,000 psi as described for IroT (Microfluidics M-110P). Lysed cells were centrifuged to remove the cell debris (20 min, 4 °C, 20,000 × g; Thermo Scientific Sorvall LYNX 6000 centrifuge). The membrane fraction was collected by subsequent ultracentrifugation of the supernatant (1 h, 4 °C, 205,100 × g; Beckman Optima XPN80). The membrane fraction was isolated and suspended in a resuspension buffer (20 mM Tris/HCl pH = 8, 500 mM NaCl, 1% (w/v) glycerol supplemented with EDTA-free protease inhibitor cocktail) to obtain a final of 1 g of cells per 1 mL of buffer. Membrane fractions were flash-frozen and stored at −80 °C until purification.

The purification of CopA was performed by affinity chromatography through the N-term Strep-tag (II) on a 5 mL Strep Trap affinity column connected to an AKTA Pure FPLC purification system (GE Healthcare). For a typical purification, CopA protein was extracted from 5 mL of membrane suspension with 1% *n*-Dodecyl- ß-D-maltoside (DDM) (Anatrace) in ice-chilled extraction buffer (20 mM Tris/HCl pH = 8, 500 mM NaCl, 1 mM EDTA pH = 8, 5 mM β-mercaptoethanol, 1% (w/v) DDM and EDTA-free protease inhibitor cocktail (Thermo Scientific)). Unextracted proteins and residual membranes were removed by ultracentrifugation (20 min, 4 °C, 205,100 × g; Beckman Optima XPN80). The detergent-solubilized protein was purified using affinity chromatography by loading to a Strep Trap column equilibrated in wash buffer (20 mM Tris/HCl pH = 8, 500 mM NaCl, 1 mM EDTA pH = 8, 1 mM dithiothreitol (DTT), 0.05% (w/v) DDM) using an AKTA Pure FPLC system (GE Healthcare). Unbound protein impurities were removed by washing the column with 20 CV wash buffer. The CopA protein was eluted with 6 CV of elution buffer containing 2.5 mM D-desthiobiotin (20 mM Tris/HCl pH = 8, 500 mM NaCl, 1 mM EDTA pH = 8, 1 mM dithiothreitol (DTT), 0.05% (w/v) DDM and 2.5 mM D-desthiobiotin (Iba solutions)). Eluted CopA protein was immediately rebuffered in desalting buffer (20 mM MOPS/NaOH pH = 7, 500 mM NaCl, 1 mM DTT, 0.05% (w/v) DDM), by loading on a preequilibrated HiPrep 26/10 desalting column (GE Healthcare). Purified protein was concentrated to ~6 mg mL$^{-1}$ using 100 KDa molecular weight cut-off (MWCO) filters (Sartorious VIVASPIN 20) by centrifugation (2100 × g, 4 °C; Thermo scientific Sorvall ST8 centrifuge). SEC was carried out on a preequilibrated Superdex 200 10/300 column (GE Healthcare) with the desalting buffer to remove any aggregated proteins and impurities. Concentration of the purified CopA protein was determined by measuring the Abs$_{280}$ (ε = 70,275 M$^{-1}$ cm$^{-1}$) or by using SDS-PAGE densitometry analysis. Freshly purified protein was reconstituted in proteoliposomes without freezing.

**Functional reconstitution of CopA in proteoliposome SUVs.** Incorporation of purified CopA protein in proteoliposomes was performed similarly to IroT using commercially available *E. coli* polar lipid extract (Avanti Polar Lipids) and L-α-phosphatidylcholine (from chicken egg; Avanti Polar Lipids). CopA was reconstituted at a 1:25 protein-to-lipid ratio of (w/w). *E. coli* polar lipids and L-α-phosphatidylcholine were initially mixed at a 3:1 ratio (w/w). A thin film of lipid was prepared from lipids dissolved in chloroform in a pear-shaped flask by flushing the mixture under a $N_2$ stream to completely remove the solvent. The lipid film was further dried overnight by storing in a vacuum desiccator and resuspended in 1 mM DTT in $H_2O$ (treated with Chelex (Bio-Rad) to remove metal contaminations). The lipid suspension was buffered to final lipid concentration of 25 mg mL$^{-1}$ with the buffer (20 mM MOPS/NaOH pH = 7, 100 mM NaCl, 1 mM DTT) using a 10× buffer stock treated with Chelex resin. SUVs were prepared by subjecting the lipid suspension to 3 freeze–thaw cycles and subsequent extrusions through PC membranes with a 1 mL gastight syringe system (Avanti, Polar Lipids, Inc). 11 extrusions were performed sequentially using PC membranes of decreasing pore sizes as described for IroT. 0.02% (w/v) DDM was added to destabilize the prepared SUVs and tilted for 1 h at 25 °C. Purified CopA stocks (~6 mg/mL) were added to the detergent destabilized liposomes on ice to a final 1:25 (w/w) protein:lipid ratio. Detergent was removed by adding Bio-Beads (SM-2; Bio-Rad), activated as described above. Complete detergent removal was attained by exchanging Bio-beads in proteoliposome solution after 1, 2, 16, and 18 h by tilting at 4 °C. Proteoliposome pellets were collected by ultracentrifugation in a Sorvall mX120+ Micro-Ultracentrifuge (45 min, 4 °C, 160,000 × g). Proteoliposome pellets resuspended in 20 mM MOPS/NaOH pH = 7, 100 mM NaCl, 1 mM DTT (treated with Chelex) to a final lipid concentration of 25 mg mL$^{-1}$ (CopA = 1 mg mL$^{-1}$). Protein incorporation in SUVs was determined by analyzing the supernatant and the resuspended pellets using SDS-PAGE (4–15% Tris-Glycine Mini-PROTEAN gels, BioRad).

**Determination of specific ATPase activity of CopA-detergent micelles encapsulated in ZIF upon exfoliation.** CopA protein immobilized in ZIF, stressed and subsequently exfoliated was buffer exchanged to remove EDTA by passing through a 5 mL HiTrap desalting column (GE Healthcare). ATPase assays were performed using solutions prepared in Chelex treated MiliQ water. Protein concentration upon exfoliation was determined by SDS-PAGE densitometric analysis prior to the assay. CopA solutions (34.4 μL) were placed in separate wells of a 96-well plate. 1 M MgCl$_2$ and 100 mM Cysteine were added separately in the wells to obtain final concentrations as 10 mM MgCl$_2$ and 1 mM cysteine. CuCl$_2$ (5 μM–2 mM) was added to separate wells to obtain final Cu$^+$ concentration of 0.05 uM–20 μM. 10 mM ATP stock (to final 1 mM) was added in each well to initiate the reaction. The 96-well plate carrying the reaction mixtures was incubated at 37 °C for 20 min under shaking (350 rpm; Eppendorf ThermoMixer). The reaction was stopped by addition to each well at the same time of 10 μL of working reagent from the Malachite green-phosphate assay kit, (Sigma-Aldrich MAK307) containing Malachite green. The reaction mixture was incubated at RT for 10 min for color development and the absorbance measured at 620 nm using a Tecan Spark 20 M plate reader. Controls were carried out in the absence of MgCl$_2$ using a saturating concentration of Cu$^+$ (10 μM final). The inorganic phosphate generated was determined by using a standard curve for inorganic phosphate (P$_i$) measured in parallel with the ATPase assays. The specific activity of CopA was calculated as nmol of inorganic phosphate generated (nmol P$_i$ mg$^{-1}$ min$^{-1}$). ATPase assays were carried out for the protein encapsulated in ZIF and stressed with different temperatures (25, 55, and 80 °C) or protein denaturant (0.1% SDS (w/v)).

**Determination of specific ATPase activity of CopA proteoliposomes immobilized in ZIF upon exfoliation.** CopA proteoliposomes were extruded sequentially through 1 μM, 0.4 μM and 0.2 μM membrane filters using gastight syringes prior to the encapsulation in ZIF. The buffer upon CopA proteoliposome exfoliation was exchanged by ultracentrifugation in a Sorvall mX120+ Micro-Ultracentrifuge (45 min, 4 °C, 160,000 × g). The proteoliposome pellets were resuspended in 20 mM MOPS pH = 7, 100 mM NaCl, and 1 mM DTT. Reaction mixtures, prepared for the assays in Eppendorf tubes similarly as described for detergent-solubilized CopA, were incubated at 37 °C for 20 min under agitation (350 rpm, Eppendorf ThermoMixer). After completion of the reaction, the mixtures were centrifuged (3 min, 4 °C, 160,000 × g) in a Sorvall mX120+ Micro-Ultracentrifuge to remove aggregated lipids. 40 μL samples were placed in separate wells in a 96-well plate. Malachite green reagent was added to the reaction mixtures and absorbance measured at 620 nm after allowing for color development at RT as described for detergent-solubilized protein and specific activity determined.

**Preparation of Lp@ZIF.** Two metal-to-ligand ratios were used for the encapsulation of blank liposomes. The lower ratio (20 × 16) encapsulation was performed as follows: 40 μL of a 12.5 mg mL$^{-1}$ stock solution of blank liposome was added to a mixture of 4326 μL of M-buffer (20 mM MOPS, 100 mM NaCl, and 1 mM TCEP pH = 7) and 534 μL of 3 M 2-methylimidazole. After thoroughly mixing, 100 μL of 1 M zinc acetate dihydrate were added to the solution. Immediately after adding the metal, the mixture went from colorless to cloudy. The solution was incubated for 18 h at RT and centrifuged at 2071 × g for 15 min. The resulting pellet was washed twice with ultrapure water. After washing, the pellet was either left to dry at RT or

used as is. Accordingly, controls consisting of pristine ZIF were prepared by combining 4366 μL M-buffer (20 mM MOPS, 100 mM NaCl, and 1 mM TCEP pH = 7), 534 μL of 3 M 2-methylimidazole, and 100 μL of 1 M zinc acetate dihydrate. The mixture was incubated for 18 h at RT, centrifuged at 2071 × g for 15 min and the supernatant collected for further characterization. The ZIF pellet was washed twice with ultrapure water, and either dried at RT or used as is. For the higher metal to ligand ratio (40 × 16) 40 μL of a 12.5 mg mL⁻¹ blank liposome was added to a solution consisting of 3,693 μL M-buffer (20 mM MOPS, 100 mM NaCl, and 1 mM TCEP pH = 7) and 1,067 μL of 3 M 2-methylimidazole. Once thoroughly mixed, 200 μL of 1 M zinc acetate dihydrate were added to the colorless solution. Upon interaction of the metal with the mixture, the solution went from colorless to cloudy. Further, the encapsulated liposomes were incubated for 18 h at RT. The pellet was harvested by centrifugation at 2071 × g for 15 min and the supernatant collected for encapsulation efficiency determination. Similarly, pristine ZIF controls were prepared by mixing 3733 μL M-buffer (20 mM MOPS, 100 mM NaCl, and 1 mM TCEP pH = 7), 1067 μL of 3 M 2-methylimidazole, and 200 μL of 1 M zinc acetate dihydrate. The mixture was incubated for 18 h at RT, centrifuged at 2071 × g for 15 min and the supernatant used for characterization. The ZIF pellet was washed with ultrapure water, and either used as is or dried at RT.

**Preparation of Lp@ZIF-8**. Three conditions were further investigated (20 × 32, 20 × 64, and 40 × 32). The 20 × 32 Lp@ZIF-8 was prepared as follows: 758 μL of M-Buffer, 214 μL of 3 M MIM, and 8 μL of 12.5 mg mL⁻¹ liposome were placed in a 1 mL Eppendorf vial. Then, 20 μL of 1 M zinc acetate dihydrate was added to the mixture. Similarly, the 20 × 32 ZIF-8 control was prepared by reacting 766 μL of M-Buffer, 214 μL of 3 M MIM, and 20 μL of 1 M zinc acetate dihydrate. Next, the 20 × 64 Lp@ZIF-8 sample was prepared by mixing 545 μL of M-Buffer, 427 μL of 3 M MIM, 8 μL of 12.5 mg mL⁻¹ liposome, and 20 μL of 1 M zinc acetate dihydrate. Further, pristine 20 × 64 ZIF-8 was synthetized upon incubation of 553 μL of M-Buffer, 427 μL of 3 M MIM, and 20 μL of 1 M zinc acetate dihydrate. Finally, the 40 × 32 Lp@ZIF-8 composite was afforded by mixing 525 μL of M-Buffer, 427 μL of 3 M MIM, 8 μL of 12.5 mg mL⁻¹ liposome, and 40 μL of 1 M zinc acetate dihydrate. Whereas the 40 × 32 ZIF-8 control was made by reacting 533 μL of M-Buffer, 427 μL of 3 M MIM, and 40 μL of 1 M zinc acetate dihydrate. All reactions followed a 4 h incubation at room temperature. Excess precursors were removed via centrifugation at 1360 × g for 10 min and washed twice with water. Exfoliation of the liposomes was done through composite incubation in a solution consisting of 0.05 M EDTA, pH 7 suspended in M-Buffer.

**Lp@ZIF temperature stressing**. Several batches of both 20 × 16 and 40 × 16 was prepared and separated as follows: RT stressed, 55 °C stressed, 80 °C stressed, and 20-day stressed. Controls for each of the stressing conditions included pristine (non-stressed, freshly extruded) and nonencapsulated stressed liposomes. The stressing at RT was performed by leaving the samples in the bench top for 48 h. The 55 °C stressed samples were placed inside a 500 μL plastic tube and incubated in a thermocycler for 15 min. Similarly, the 80 °C stressing experiment proceeded by incubating the samples in the thermocycler for 5 min. Further, the 20-day stressing experiment consisted of leaving the samples in the bench for the aforementioned time. After stressing, all samples were exfoliated in a solution consisting of 20 mM MOPS, 100 mM NaCl, 1 mM TCEP, and 0.05 M EDTA pH = 7.5. Exfoliated samples were then immediately characterized under TEM and DLS.

**Preparation of IroT@ZIF**. For the immobilization of IroT only the higher ratio (40 × 16) exhibited favorable results after encapsulation and stressing. On a typical experiment, 625 μL of 2 mg mL⁻¹ IroT stock was combined with 3108 μL M-buffer (20 mM MOPS, 100 mM NaCl, and 1 mM TCEP pH = 7) supplemented with CYMAL-7 (0.05% (w/v)) and 1,067 μL of 3 M 2-methylimidazole. The resulting aqueous solution was thoroughly mixed, and 200 μL of 1 M zinc acetate dihydrate added. Upon addition of the zinc solution, the former colorless aqueous mixture turned into a milky suspension. The suspension was left in the bench for 18 h at RT, centrifuged at 2071 × g for 15 min and the supernatant was used for encapsulation efficiency quantification. The resulting pellet was washed twice with water to remove unreacted precursors and either dried at RT or used as is. Similarly, pristine ZIF controls were made by reacting 3,733 μL M-buffer (20 mM MOPS, 100 mM NaCl, and 1 mM TCEP pH = 7) supplemented with CYMAL-7 (0.05% (w/v)), 1,067 μL of 3 M 2-methylimidazole, and 200 μL of 1 M zinc acetate dihydrate. The cloudy suspension was aged at RT for 18 h, centrifuged at 2071 × g for 15 min and the supernatant kept for further characterization. The ZIF pellet was washed twice with ultrapure water and used as is.

**IroT@ZIF temperature stressing**. Similar to the Lp@ZIF stressing, several batches of 40 × 16 were synthetized and separated under the following categories: RT, 55 °C, and 80 °C. Controls for each stressing condition included −80 °C refrigerated and nonencapsulated IroT. Stressing at RT was done by leaving the samples on the bench top for 48 h. The 55 °C stressing experiment proceeded by incubating dried samples in a thermocycler for 15 min. Further, stressing at 80 °C proceeded by incubating the samples in the thermocycler for 5 min. SEC characterization of the stressed samples was followed after exfoliation of ZIF-8 immobilized IroT samples using an aqueous mixture of 20 mM MOPS, 100 mM NaCl, 1 mM TCEP, 0.05 M EDTA pH = 7.5, and CYMAL-7 (0.05% (w/v)).

**IroT@ZIF stressing in denaturant agents**. A batch of freshly prepared IroT@ZIF was incubated in 0.1% sodium-dodecyl sulfate (SDS) for 30 min, and thoroughly washed five times with ultrapure water. The pellet was harvested upon centrifugation at 2071 × g for 15 min, immediately exfoliated using a mixture of 20 mM MOPS, 100 mM NaCl, 1 mM TCEP, 0.05 M EDTA pH = 7.5, and CYMAL-7 (0.05% (w/v)) and characterized through SEC. Controls included non-stressed pristine −80 °C refrigerated and non-encapsulated stressed IroT.

**Preparation of CopA@ZIF**. Two concentration ratios of zinc acetate dihydrate and 2-methyl imidazole were used (20 × 16 and 40 × 16). For the 20 × 16 condition, 1000 μL of 0.5 mg mL⁻¹ CopA stock was combined with 533.3 μL of 3 M 2-methyl imidazole and 3370 μL of 20 mM MOPS, 100 mM NaCl, 1 mM DTT pH = 7 and 0.05% (w/v) DDM. The resulting aqueous solution was thoroughly mixed, and 100 μL of 1 M zinc acetate dihydrate was added. For the 40 × 16 ratio, 1000 μL of 0.5 mg mL⁻¹ stock solution of purified CopA was combined with 1066.7 μL of 3 M 2-methyl imidazole and 2733 μL of 20 mM MOPS, 100 mM NaCl, 1 mM DTT pH = 7 and 0.05% (w/v) DDM. The resulting aqueous solution was thoroughly mixed, and 200 μL of 1 M zinc acetate dihydrate was added. CopA@ZIF crystals were harvested by centrifugation at 2071 × g for 15 min. The obtained crystals were washed twice with ultrapure water. The resulting bio-composites were kept as a suspension by leaving a layer of ultrapure water (~200 μL). Supernatants from first wash were collected and used for determination of encapsulation efficiency. Similarly, 20 × 16 pristine ZIF control was prepared by reacting 4366.7 μL of 20 mM MOPS, 100 mM NaCl, 1 mM DTT pH = 7 and 0.05% (w/v) DDM, 533.3 μL of 3 M 2-methylimidazole, and 100 μL of 1 M zinc acetate dihydrate. Similarly, 40 × 16 pristine ZIF controls were made by reacting 3733.3 μL of 20 mM MOPS, 100 mM NaCl, 1 mM DTT pH = 7 and 0.05% (w/v) DDM, 1066.7 μL of 3 M 2-methylimidazole, and 200 μL of 1 M zinc acetate dihydrate. The cloudy suspension was aged at RT for 18 h, centrifuged at 2071 × g for 15 min and the supernatant kept for further characterization. The ZIF pellet was washed twice with ultrapure water and used as is.

**CopA@ZIF temperature stressing**. CopA@ZIF was stressed under three different temperatures following the procedure from IroT above. Several batches of 20 × 16 and 40 × 16 were freshly prepared. After stressing, samples were exfoliated using 0.05 M EDTA solution containing 20 mM MOPS, 100 mM NaCl, 1 mM DTT pH = 7 and 0.05% (w/v) DDM. Exfoliated CopA was desalted using a 5 mL HiTrap column (GE Healthcare). Following desalting, ATP hydrolysis activity was determined for each of the tested samples.

**CopA@ZIF stressing in denaturant agents**. Identically to IroT, CopA@ZIF biocomposites were stressed with 0.1% SDS. After incubation in the SDS solution for 30 min, crystals were harvested by centrifugation at (2071 × g for 15 min, washed five times with ultrapure water and immediately exfoliated with an 0.05 M EDTA solution containing 20 mM MOPS, 100 mM NaCl, 1 mM DTT pH = 7 and 0.05% (w/v) DDM. Exfoliated CopA was desalted using a 5 mL HiTrap column (GE Healthcare). Finally, all tested CopA samples were characterized through ATP-hydrolysis activity assays.

**Preparation of IroTPL@ZIF proteoliposome bio-composites**. 500 μL of 12.5 mg mL⁻¹ IroT proteoliposome stock were combined with 246 μL M-buffer (20 mM MOPS, 100 mM NaCl, and 1 mM TCEP pH = 7), and 214 μL of 3 M 2-methylimidazole. After carefully mixing, 40 μL of 1 M zinc acetate dihydrate was added. Immediately after addition of the zinc solution, a milky suspension was obtained. The reaction was aged at RT for 18 h, centrifuged at 2071 × g for 15 min, washed twice with water and twice with methanol to remove unreacted precursors and either used as is or left drying at RT.

**IroTPL@ZIF temperature stressing**. The experiments for stressing follow the same parameters as the ones designed for blank liposomes and IroT as detailed above. Separate freshly prepared samples were stressed under different temperature conditions, exfoliated, and characterized under DLS and TEM.

**Preparation of CopA-PL@ZIF proteoliposome bio-composites**. CopA-PL@ZIF was synthesized according to the same procedure shown for the synthesis of CopA@ZIF bio-composites. 500 μL of 12.5 mg mL⁻¹ CopA proteoliposome stock were combined with 373.3 μL of buffer (20 mM MOPS, 100 mM NaCl, and 1 mM DTT pH = 7) and 106.7 μL of 3 M 2-methylimidazole. After carefully mixing, 20 μL of 1 M zinc acetate dihydrate were added. Immediately after addition of the zinc solution, a milky suspension was obtained. The reaction was incubated at RT for 18 h, centrifuged at 2071 × g for 15 min, and washed twice with ultrapure water. Bio-composites were either left to dry at RT or used immediately for the stressing experiments.

**CopA-PL@ZIF temperature stressing**. Freshly extruded CopA proteoliposomes were encapsulated as formerly mentioned and used for temperature stressing following identical experimental guidelines as for blank liposomes and IroT detergent–protein micelles discussed above. After stressing, samples were immediately exfoliated and characterized by performing ATP-hydrolysis activity assays.

**Cy5 liposome encapsulation**. Cy5 fluorescence dye was encapsulated into liposomes prepared with *E. coli* polar lipids and L-α-phosphatidylcholine 3:1 ratio (w/w) (Avanti Polar Lipids). Liposomes were diluted to 12.5 mg mL$^{-1}$ final lipid concentration with M-buffer (20 mM MOPS, 100 mM NaCl, and 1 mM TCEP pH = 7). Cy5 was added to a final concentration of 200 nM and encapsulated to the liposome lumen by freeze–thaw membrane fracture followed by extrusion through PC membranes with decreasing pore sizes (1 μm, 400 nm, and 200 nm), similar to the liposome preparation protocol. Excess Cy5 was removed by three successive ultracentrifugation and resuspension steps with M-buffer at 160,000 × g at 4 °C in a Sorvall mX120+ Micro-Ultracentrifuge for 45 min. The Cy5-encapsulated liposome mixture was finally resuspended to the initial volume with the M-buffer for ZIF immobilization

**Lp@ZIF encapsulation efficiency determination**. Both 20 × 16 and 40 × 16 Lp@ZIF composites were prepared for determination of encapsulation efficiency. Briefly, the 20 × 16 Lp@ZIF composite was prepared by mixing 4326 μL of M-Buffer, 534 μL of 3 M MIM, 40 μL of 12.5 mg mL$^{-1}$. Then, 100 μL of 1 M zinc acetate dihydrate was added to the mixture. On the other hand, the 40 × 16 Lp@ZIF composite was prepared by 3693 μL of M-Buffer, 1067 μL of 3 M MIM, 40 μL of 12.5 mg mL$^{-1}$. Further, 200 μL of 1 M zinc acetate dihydrate was mixed with into the solution. Composites were then allowed to sit at RT for 18 h. Supernatants were collected upon centrifugation at 4000 × g for 10 min. Emission spectra were then measured on the collected supernatants. Samples were placed in a sub-micro quartz cell (Starna Cells) and emission spectra collected from 640–700 nm with 1 nm increments at 25 °C in a spectro-fluorometer (Horiba scientific Fluoromax-4) at an excitation wavelength of 620 nm, using an excitation and emission slit width of 5 nm.

**Cationic liposome preparation**. For liposome preparation with cationic lipids possessing positive polar heads, lipid composition was selected as follows: *E. coli* polar lipids: 1,2-dioleoyl-3-trimethylammonium propane (DOTAP): L-α-phosphatidylcholine in a 1:2:1 ratio (w/w) (Avanti Polar Lipids). Liposomes were prepared following the same protocol as for liposomes with negative polar heads.

**Zn$^{2+}$ liposome binding assay**. Liposome samples were diluted to 12.5 mg mL$^{-1}$ final lipid concentrations with the M-buffer and freshly extruded through PC membranes with decreasing pore sizes (1 μm, 400 nm, and 200 nm) prior to the experiment followed by three freeze–thaw cycles. Each liposome sample was separately treated with 1 M Zn(CH$_3$CO$_2$)$_2$ stock solution to set the final Zn$^{2+}$ concentration to 20 mM and 40 mM and incubated at RT for 1 h to allow Zn$^{2+}$ binding, After incubation, unbound Zn$^{2+}$ was removed by three successive rounds of ultracentrifugation and resuspended with M-buffer steps (160,000 × g, 4 °C, 45 min) in a Sorvall mX120+ Micro-Ultracentrifuge. Zn$^{2+}$ bound liposomes were digested with 35% HNO$_3$ for 48 h. Samples were then diluted to 2% HNO$_3$ and Zn$^{2+}$ bound to the liposome measured with inductively-coupled plasma mass spectrometry (ICP-MS).

**Material characterization**. Surface morphology was investigated using a Zeiss Supra 40 scanning electron microscope at 3 kV and a working distance of 6–10 mm. Each sample was sputtered with ~40 Å layer of gold. Proteoliposomes and blank liposomes were observed before and after encapsulation on a JEOL JEM-1400 plus transmission-electron microscope at 120 kV.

Surface area measurements were carried out on a Micrometrics ASAP 2020 surface area analyzer by nitrogen adsorption under 77 K. Surface areas were assessed through Brunauer–Emmett–Teller (BET) method and pore sizes calculated by a non-localized density functional theory with a carbon slit pore model[18]. Samples were activated prior to surface area measurements by soaking with MeOH and drying them under vacuum for 24 h. The MeOH solution was decanted and replaced with dichloromethane. Samples were dried again under vacuum for an additional 24 h. Prior to analysis samples were degassed at 120 °C under vacuum for 12 h.

PXRD data was measured using a Rigaku SmartLab X-ray diffractometer equipped CuKα (1.54060 Å) at 30 mA and 40 kV. Samples were activated through MeOH and DCM soaking, dried under vacuum for 24 h, and degassed under N$_2$ for 12 h. Phase identification was done using the ZIF phase analysis software developed by Carraro et al. Data for each individual PXRD collected was uploaded into the software and analyzed from 6° to 39° (2θ)[56].

Dynamic-light scattering was measured with a Malvern Panalytical Zetasizer Nano ZS. Measurements were collected using a disposable cuvette at RT with a 633 nm laser source, a medium refractive index of 1.33, a material refractive index of 1.51, and a scattering angle of 175°.

TGA thermal stabilities were tested using a TA Instruments SDT Q600 Analyzer, from 30 to 800 °C, under a N$_2$ atmosphere, with a heating rate of 5 °C min$^{-1}$.

**Sample shipping**. Samples were pipetted into sturdy glass vial into a single bubble padded USPS mailer and shipped via ground service from Dallas, Texas to a recipient in the state of Rhode Island, USA. Upon arrival and subsequent storage for two months at RT, the proteoliposomes encapsulated in ZIF-L were exfoliated and ATPase assays were carried out on the proteoliposomes rebuffered in M buffer (20 mM MOPS pH = 7, 100 mM NaCl, 1 mM DTT), after ultracentrifugation (Sorvall mX120+ Microcentrifuge, min, 4 °C, 160,000 × g) to remove EDTA. Control assays were done on the shipped and exfoliated proteoliposomes.

**Time-resolved small-angle X-ray scattering (SAXS)**. Time-resolved SAXS have been collected at the SAXS beamline (ELETTRA synchrotron light source)[57]. We operated at photon energy of 8 keV covering the range of momentum transfer, $q = 4\pi \sin(\theta)/\lambda$, between 0.1 and 7.2 nm$^{-1}$. We monitored the kinetics of the nucleation and growth of MOF particles by using a commercial stopped flow apparatus (SF4, Bio-Logic, Grenoble, France) especially designed for Synchrotron Radiation SAXS investigations. Three independently driven syringes were filled respectively with the Zn$^{2+}$, the 2-methylimidazole and the liposome solutions. With three step-motors we controlled the volumes mixed in a micromixer that was subsequently injected in a horizontal 1 mm quartz capillary placed in the X-Ray beam (Supplementary Fig. 8A). The volume ratio between the three solutions was set to maintain the final concentrations used for the syntheses in batch.

For each experiment, a total volume of 1 mL was injected in the micromixer and then, the solution was transferred to a 1 mm capillary horizontally oriented (Supplementary Fig. 8A). The start of the mixing sequence is triggered from the X-ray data-acquisition system, which took images with a time resolution of 100 ms (detector: Pilatus3 1M, Dectris Ltd, Baden, Switzerland; sample to detector distance: 1260 mm, as determined with a silver behenate calibration sample). All the experiments were performed at RT. The ligand solution was measured to record the background that was subtracted from the data sets. The resulting two-dimensional images were radially integrated to obtain a 1D pattern of normalized intensity versus scattering vector q[58]. With this set-up, we investigated the nucleation and growth of the samples 20 × 16 Lp@ZIF, 20 × 16 ZIF, 40 × 16 Lp@ZIF, 40 × 16 ZIF and the crystallization kinetics of 40 × 16 Lp@ZIF, 40 × 16 ZIF.

The crystallization kinetics of the samples 20 × 16 Lp@ZIF and 20 × 16 control were investigated by manually filling the precursors into a vertically positioned glass capillary (Supplementary Fig. 8B). This was necessary because of the rapid flocculation of the particles that inhibit their detection in the horizontally mounted stop-flow capillary (Supplementary Fig. 8C). The precursors (total volume: 1 ml) were pre-mixed in an Eppendorf tube and mixed following the same procedure described for the preparation of the batch samples. After mixing the precursors, the solution was transferred in a 1.5 mm capillary using a syringe. Then, the capillary was sealed with parafilm and mounted vertically on the SAXS sample holder. The measurement started 120 s after the mixing of the precursors. Every 8 s, a SAXS pattern was collected and three different vertical positions of the capillary were investigated over time. This means that a specific vertical position was measured every 24 s. The scan over three different positions were performed to simultaneously investigate the bulk of the solution and the precipitate that accumulates at the bottom of the capillary over time.

Data analysis was performed using the software package Igor Pro (IGOR Pro 7.0.8.1, Wavemetrics, USA). The Invariant $\tilde{Q}$ is related to the Porod invariant of the scattering curve and is defined as:

$$\tilde{Q} = \int_{q_1}^{q_2} dq * q^2 * I(q), \tag{1}$$

where $q_1 = 0.1$ nm$^{-1}$ and $q_2 = 0.6$ nm$^{-1}$[59].

The increase of $\tilde{Q}$ over time describes the formation of agglomerates/particles within the investigate volume. A plateau in the time series of $\tilde{Q}$ values, indicates stationary conditions. To evaluate the morphology of the growing particles, the SAXS patterns were fitted. For the plate-like morphology, a simplified model for quasi-infinite large plates with the thickness T was used. This model is based on the Cauchy integral[60] and the asymptotic behavior of the scattering being proportional to $1/q^2$ in the q-range $q \ll 1/\xi$ due to the large (infinite) lateral size of the plate-like particles. The form factor $P(q)$ is:

$$P(q) = \frac{1}{(1 + \xi^2 q^2) q^2} \tag{2}$$

with ξ as thickness correlation parameter.

ξ can be related to plate thickness T with $T = \sqrt{12}\xi$, if compared with the Guinier approximation for quasi-infinite plates. The integrated scattering intensity $I(200)$ is defined as:

$$I(200) = \int_{q_1}^{q_2} dq * I(q), \tag{3}$$

where $q_1 = 5$ nm$^{-1}$ and $q_2 = 5.4$ nm$^{-1}$.

## Data availability

Data supporting the results of this work are available from the corresponding authors upon reasonable request.

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

## Acknowledgements

This project was partially funded by The University of Texas at Dallas Office of Research through the SPIRe grant program. J.J.G. thanks the National Science Foundation (CAREER DMR-1654405 and DMR-2003534) and the Welch Foundation (AT-1989- 20190330). The work in G.M. laboratory was supported by the Robert A. Welch Foundation (Grant: AT-1935-20170325 to G.M.) and by the National Institute of General Medical Sciences of the National Institutes of Health under Award Number R35GM128704 (to G.M.). The content is solely the responsibility of the authors and does not necessarily represent the official views of the National Institutes of Health. R.A.S. acknowledges support from the Army Research Laboratory (W911NF-18-2-0035). A.D.S. acknowledges Consejo Nacional de Ciencia y Tecnología (National Council for Science and Technology) of Mexico for a doctoral fellowship. F.C. and P.F. acknowledge European Union's Horizon 2020 Program (FP/2104-2020)/ERC Grant Agreement no. 771834 POPCRYSTAL and the CERIC-ERIC Consortium for the access to experimental facilities and financial support.

## Author contributions

F.C.H., S.S.A, and N.S.A. contributed equally to this paper. Primary paper writing and editing was done by F.C.H., G.M., and J.J.G. Blank liposome, IroT, and IroTPL encapsulation, stressing, was done by F.C.H. IroT expression, purification, proteoliposome preparation, SECs of IroT/IroTPL were performed by S.S.A. CopA expression purification, proteoliposome preparation, and ATPase activity assays were done by N.S.A. CopA/CopA-PL encapsulation and stressing were done by Y.H.W. SDS was done by F.C.H., S.S.A., N.S.A., and Y.H.W. DLS was done by F.C.H., S.S.A., and Y.H.W. TEMs were taken by O.R.B. SEMs were taken by M.A.L. and O.R.B. PXRDs were done by F.C.H. and M.A.L. TGA was done by A.D.S. Nitrogen sorption and BET analysis were done by S.D.D. The synthesis of Lp@ZIF was independently verified and SAXS/WAXS experiments were performed by F.C., H.A., and P.F.; SAXS/WAXS data was interpreted by F.C., H.A., and P. F. Funding was raised by R.A.S, G.M., and J.J.G.

## Competing interests

The authors declare no competing interest.
