## [Peer Review File · Nature Communications]

REVIEWER COMMENTS

Reviewer #1 (Remarks to the Author):

The manuscript titled "Stabilization of supermolecule membrane protein lipid bilayer assemblies through immobilization in a crystalline exoskeleton" described the protection of liposome, transmembrane protein and proteoliposomes through ZIF-8. A special buffer, called M buffer, was employed as the medium to stabilize the encapsulated species during the encapsulation process. The authors need to address several concerns noted below before this paper can be published.

1. Liposomes are recommended to be stored in 4°C and can be stable for days. Here some of the comparison for size and shape of liposome were conducted at RT for 48 hour (page 4, line 3). How do the ZIF-8 protected liposome compare to those stored at 4°C? Same question also applies to enzyme activity in transmembrane protein and proteoliposome sections, where the control samples (without encapsulation) are stored at RT not at recommended condition.

2. Are the conditions of ZIF-8 encapsulation same for all three encapsulated species? If they are the same, why the pristine ZIF-8 crystals prepared in liposome section and transmembrane section different, as shown by PXRD in Figure 1B and 2F (both named 40X16 ZIF-8)? If they are different, please be more specific about the method.

3. To demonstrate the protection efficacy, the liposome@ZIF-8 is stored in powder state compared to liposome in solution state at same temperature. Is this a fair comparison? Is this same in the other two cases? Although liposome in dry powder is hard to prepare, it is feasible to prepare liposome@ZIF-8 in solution. It will be better to compare them at same state.

4. In liposome part, the author measured the size and shape of liposome with and without ZIF-8 protection. However, it is also important to indicate how much liposome has been encapsulated. What is the encapsulation efficiency? What fraction of the liposomes are lost?

5. In Figure 3I, red legend represents "CopA stressed". Should it be "CopAPL stressed"? Otherwise, there is no meaning to compare with "20X16 CopAPL@ZIF-8".

Reviewer #2 (Remarks to the Author):

In this study, Herbert et al. report the stabilization of liposomes, membrane proteins and proteoliposomes through encapsulation in ZIF-8. One of the technical challenges associated with the study of transmembrane proteins the lack of stability of such proteins during detergent-based purification and their long-term storage. As the authors nicely articulate in the Introduction, different approaches have been developed to stabilize membrane proteins in a native-like state (e.g., through the use of nanodiscs or amphipols), but all suffer from certain drawbacks. Now, following up on the many studies in which MOF's have been used to encapsulate and stabilize soluble proteins, the authors apply a similar strategy to membrane-based systems.

The topic of the paper is timely and the study nicely combines the areas of expertise of three research groups. In general, I thought that the paper was put together relatively well (with some exceptions – see below) and the main findings were supported by the results. However, I found the paper to be "observational" rather than "insightful" and it is the latter attribute that distinguishes a great study from a good or interesting one. While I can see why the ZIF-mediated stabilization of membrane proteins/proteoliposomes would be more than an incremental advance over other MOF-protein stabilization studies in the literature, I feel that some of the authors' observations on the morphologies and properties of the ZIFs around the lipid and proteolipid constructs generate more questions than answers. So, I am on the fence as to whether the manuscript in its current form is appropriate for Nature Communications or not.

-In my opinion, the most distinct aspect of this study from others in the field is the interactions between ZIF-8 coat and the lipid environment. How do these interactions affect the formation of different ZIF-8 lattice structures? What are the kinetics of formation of the ZIF-8 coat? How are they affected by the solution conditions (pH, temperature, ionic strength, additives)? The authors report only two conditions (20 x 16 and 40 x 16) for ZIF-8 formation, which doesn't cover a large screening space, yet surprisingly, even these conditions give rise to substantially different results (framework structure, porosity, stability etc.). But to me, it is not clear why only such a narrow window of conditions works well or is reported, why such large differences between the two conditions are observed and how the nature of the lipids or proteins affects ZIF-8 formation? Only with insights that result from explanations to these questions will it be possible to understand the systems in hand in better chemical detail and manipulate them properly.

- Related to the points above, the crystalline, ZIF-8 exoskeleton is not a phase-pure MOF material, which is what has been observed in other protein encapsulation studies before. The authors should provide a more detailed analysis and explanation for the PXRD spectrum in Figure 1B. Other related questions: If the new reflections peaks are formed during encapsulation in the case of the 20 x 16 ZIF-8, why does the control have the same new peaks as 20 x 16 LP @ ZIF-8? In the case of 40 x 16 ZIF-8, if it matches poorly with the known polymorphs of ZIF-8, what are the any other possible structural models? Importantly, why is the PXRD of 40 x 16 ZIF-8 in Figure 2F is so different from 40 x 16 ZIF-8 in Figure 1B? On that note, the blue and red traces in Figure 2F look "too identical".

- The TGA of the 40 x 16 LP@ZIF sample shows a gradual ~30% mass loss starting around 120 C, not 200 C (as stated in the text). If this is attributed to the decomposition of lipids, it is not clear to me as to why it is so different than what is observed in the case of the 20 x 16 LP@ZIF (especially given that the 40 x 16 ZIF-8 control is thermally more stable than the 20 x 16 ZIF-8 control).

- I don't see a correlation between porosity (or for that matter, crystallinity) and protein stabilization. So, what is then the primary determinant of protein stabilization and if there is no correlation, would it be simply ok to have an amorphous coat on the surface? Further regarding porosity: it would be interesting to see if CopA and Iro were catalytically active while encapsulated.

-It would be beneficial to directly show protein/liposome encapsulation via fluorescence microscopy.

- "aiding and abetting" is an odd term to use. It is perhaps more appropriate for a courtroom than a scientific article.

- Refs 37, 41 need to be corrected.

Reviewer #3 (Remarks to the Author):

This manuscript describes an excellent piece of research that truly advances the field of MOF biomimetic mineralisation, and, more broadly, shows how MOF-based composites can be used to stabilise proteoliposomes. The value of this result is evidenced by a series of experiments that show the composites can protect relatively fragile membrane associated proteins from stressors that typically lead to their degradation. Hitherto, biomimetic mineralisation has focused on encapsulating proteins that are far more robust and thus, this work significantly broadens the scope of this area. I think the work will be of particular interest to both materials scientists and biochemists.

I enthusiastically recommend this paper for publication in Nature Commun. as in my opinion it clearly meets the required novelty and scholarly approach required for the journal. Nevertheless, prior to acceptance there are some minor comments that I think the authors should address,

1. The authors mention, rightly, that the formation of ZIF-based biocomposites is influenced by the surface charge of the biomacromolecule (in this case the liposome). Ref 37 is incomplete, further, also the authors may want to include the following paper as it shows that the notion applies to other biomolecules *Mater. Horiz.*, 2019, 6, 969-977. In addition, did the authors measure zeta potentials? Not necessary if they don't have the data but would be useful to consider in future work

2. The isotherms shown in figure S1C are described as Type II in the manuscript. To me these are all clearly Type I. Type II isotherms have a characteristically increasing uptake which is representative of a high external:internal surface area ratio. Adsorbate condensation at saturation pressures is common where there are macro/mesoporous gaps between crystals or if large crystals are cracked. In light of this perhaps the condensation might result from a heterogeneous coating? i.e phase boundaries (ZIF-8/ZIF-C) or microcrystalline composites rather than a single crystal? The authors may wish to change their interpretation of the data for this section.

RESPONSE TO REVIEWERS

Reviewer #1 (Remarks to the Author):

The manuscript titled “Stabilization of supermolecule membrane protein lipid bilayer assemblies through immobilization in a crystalline exoskeleton” described the protection of liposome, transmembrane protein and proteoliposomes through ZIF-8. A special buffer, called M buffer, was employed as the medium to stabilize the encapsulated species during the encapsulation process. The authors need to address several concerns noted below before this paper can be published.

1. Liposomes are recommended to be stored in 4°C and can be stable for days. Here some of the comparison for size and shape of liposome were conducted at RT for 48 hour (page 4, line 3). How do the ZIF-8 protected liposome compare to those stored at 4°C? Same question also applies to enzyme activity in transmembrane protein and proteoliposome sections, where the control samples (without encapsulation) are stored at RT not at recommended condition.

We would like to thank referee 1 for their helpful remarks. To highlight the protective effectiveness in both transmembrane proteins and lipids, stability experiments were performed using proteoliposomes where the composites and controls were kept at 4°C for 48 h. Indeed, both encapsulated and non-encapsulated samples were kept in the refrigerator for 48 h. All experiments were conducted with freshly extruded proteoliposomes. CopA was used as protein model because the determination of its metal-dependent ATPase activity is an excellent quantitative proxy for testing the integrity of both liposomes and proteins embedded in the lipid bilayer in the same experiment. Structural stress of either will result in depletion of function which will reflect poor copper-stimulated ATPase activity. Tested samples were prepared as follows: First, non-encapsulated samples were carefully diluted using M-Buffer to a final concentration of 0.25 mgmL⁻¹ and immediately stored at 4°C. PL@ZIF composites were prepared by mixing M-Buffer, 2-methylimidazole, and CopA proteoliposomes (0.25 mgmL⁻¹ final concentration). After gentle mixing, a solution of zinc acetate was added to the solution. The white flocculate that formed seconds after addition of the metal precursor was immediately refrigerated. Both encapsulated and non-encapsulated CopA-PL samples were incubated at 4°C for 48 h. Time of incubation was chosen based on the observation that non-encapsulated proteoliposomes progressively loss their function with only a 10% residual activity left after 48 h. After time optimization, encapsulated samples were exfoliated using 0.05 M EDTA in M-buffer pH 7, the liposomes were collected by ultracentrifugation. The collected proteoliposomes were resuspended in M-Buffer and the ATPase activity of CopA analyzed. When compared to freshly extruded CopA-PL results show that encapsulation in ZIF-L preserved most of the ATPase activity

of CopA proteoliposomes when stored at 4°C. In contrast, a significant loss of function for the non-encapsulated proteoliposomes was observed even when kept refrigerated 4°C. The results have been included in the manuscript text and Figure S13

A)

Figure S13: CopA proteoliposome activity assessment. A) ATPase activity of exfoliated CopA-PL@ZIF after storage at 4°C for 48 h. Activity was directly compared against freshly extruded and non-encapsulated CopA-PL stored at 4°C for 48 h.

2. Are the conditions of ZIF-8 encapsulation same for all three encapsulated species? If they are the same, why the pristine ZIF-8 crystals prepared in liposome section and transmembrane section different, as shown by PXRD in Figure 1B and 2F (both named 40X16 ZIF-8)? If they are different, please be more specific about the method.

We sincerely appreciate the comments made. Metal to Ligand ratios (M:L) 20x16 and 40x16 are the same in all encapsulated species. We repeated PXRD of both liposomes and protein composites. Results show that in both cases the formation of ZIF-L phase exoskeleton takes place, a previously reported zeolitic topology with a “leaf” morphology.¹ Through detailed examination of our synthetic parameters we determined that the washing procedure, which removes solvent from the shell prior to PXRD analysis, can alter the final topology as observed in literature. An improved standardized washing protocol is now included in the Supporting Information file that affords and preserves the as-synthesized morphology. PXRD data were recollected and have now been included in the revised version of the Manuscript (Fig. 1B and 3F).

Figure 1. Characterization of artificial lipid bilayers embedded in ZIF. PXRD spectra of ZIF liposome complexes (Lp@ZIF) and ZIF controls.

Figure 3. IroT@ZIF and CopA@ZIF characterization. PXRD spectra of ZIF liposome complexes (Lp@ZIF) and ZIF controls.

3. To demonstrate the protection efficacy, the liposome@ZIF-8 is stored in powder state compared to liposome in solution state at same temperature. Is this a fair comparison? Is this same in the other two cases? Although liposome in dry powder is hard to prepare, it is feasible to prepare liposome@ZIF-8 in solution. It will be better to compare them at same state.

We are grateful for the recommendations mentioned above. Both the non-encapsulated and encapsulated composites were tested as suspensions, making them directly comparable. We have made this clearer in the main text that this was the case.

4. In liposome part, the author measured the size and shape of liposome with and without ZIF-8 protection. However, it is also important to indicate how much liposome has been encapsulated. What is the encapsulation efficiency? What fraction of the liposomes are lost?

Great suggestion. Encapsulation efficiency was address by incorporating a fluorescent dye (Cy5) into the lumen of the liposomes. Following our reported synthetic reaction conditions, encapsulation into ZIF was redone, and supernatants were collected after the first washing. Entrapment efficiency was measured using the recovered supernatant. Both tested conditions (20x16 and 40x16 Lp@ZIF) have a 90% encapsulation efficiency. Below are the spectra of the Liposome-Cy5 collected prior to encapsulation in ZIF. Data also includes fluorescence spectra recorded for both 20x16 and 40x16 Lp@ZIF supernatants. Additionally, both liposome loaded composites and controls were imaged using confocal microscopy (Cy5 channel). Obtained data shows that only the Lp@ZIF samples have fluorescence whereas the controls have no signal in the Cy5 emission channel.

Figure S1. Encapsulation efficiency determination of liposomes-embedded in ZIF-L. A) Fluorescence emission spectra of Cy5-loaded liposome versus supernatants collected for 20x16 Lp@ZIF, 40x16 Lp@ZIF. Controls include supernatants collected for both 20x16 and 40x16 ZIF-L. B) Confocal microscope caption of 20x16 and 40x16 ZIF-L. C) Confocal microscope caption of 20x16 Lp@ZIF. D) Confocal microscope caption of 40x16 Lp@ZIF. Pristine ZIF-L shows no intrinsic fluorescence when imaged under the Cy5 channel. On the other hand, both 20x16 Lp@ZIF and 40x16 Lp@ZIF prepared with Cy5-loaded liposomes have fluorescence in the cyanine channel (670 nm).

5. In Figure 3I, red legend represents “CopA stressed”. Should it be “CopAPL stressed”? Otherwise, there is no meaning to compare with “20X16 CopAPL@ZIF-8”.

We thank the reviewer of the observation. Indeed, we intended to say CopA-PL instead. Figure 4 (Figure 3 in original submission) is now properly labeled.

Reviewer #2 (Remarks to the Author):

In this study, Herbert et al. report the stabilization of liposomes, membrane proteins and proteoliposomes through encapsulation in ZIF-8. One of the technical challenges associated with the

study of transmembrane proteins the lack of stability of such proteins during detergent-based purification and their long-term storage. As the authors nicely articulate in the Introduction, different approaches have been developed to stabilize membrane proteins in a native-like state (e.g., through the use of nanodiscs or amphipols), but all suffer from certain drawbacks. Now, following up on the many studies in which MOF's have been used to encapsulate and stabilize soluble proteins, the authors apply a similar strategy to membrane-based systems. The topic of the paper is timely, and the study nicely combines the areas of expertise of three research groups. In general, I thought that the paper was put together relatively well (with some exceptions – see below) and the main findings were supported by the results. However, I found the paper to be “observational” rather than “insightful” and it is the latter attribute that distinguishes a great study from a good or interesting one. While I can see why the ZIF-mediated stabilization of membrane proteins/proteoliposomes would be more than an incremental advance over other MOF-protein stabilization studies in the literature, I feel that some of the authors' observations on the morphologies and properties of the ZIFs around the lipid and proteolipid constructs generate more questions than answers. So, I am on the fence as to whether the manuscript in its current form is appropriate for Nature Communications or not.

-In my opinion, the most distinct aspect of this study from others in the field is the interactions between ZIF-8 coat and the lipid environment. How do these interactions affect the formation of different ZIF-8 lattice structures? What are the kinetics of formation of the ZIF-8 coat? How are they affected by the solution conditions (pH, temperature, ionic strength, additives)? The authors report only two conditions (20 x 16 and 40 x16) for ZIF-8 formation, which doesn't cover a large screening space, yet surprisingly, even these conditions give rise to substantially different results (framework structure, porosity, stability etc.). But to me, it is not clear why only such a narrow window of conditions works well or is reported, why such large differences between the two conditions are observed and how the nature of the lipids or proteins affects ZIF-8 formation? Only with insights that result from explanations to these questions will it be possible to understand the systems in hand in better chemical detail and manipulate them properly.

We thank the reviewer for taking us to task. Over the previous months, we have conducted multiple experiments that paint a much more comprehensive picture of the mechanism of ZIF shell exoskeleton formation. In particular, we hypothesize that there is an interaction between Zn^{2+} and the surface of the liposomes, which templates the growth of ZIF around tested supramolecular complexes. This is a hypothesis we put forward in our work on viral nanoparticles,² and conducted the experiments below to establish this hypothesis. Secondly, we hypothesize that the formation of the ZIF on the surface is biomimetic in origin. In other words, not only does the liposome induce crystal formation, it directs its growth. Again, a hypothesis we and others have proposed for proteins and for which considerable experimentation has been conducted to demonstrate. Finally, we hypothesized that the growth of ZIF on the surface of liposomes (if, indeed, it is biomimetic in nature) can be tuned based on solution concentrations of ligand and metal. Again, we and others have proposed this for proteins and carbohydrates, and we conduct experiments to demonstrate this below.

We investigated the interactions promoting the coating of ZIF around the liposomes using ICPMS. We have previously observed that materials with even a modest association with Zn^{2+} induce crystal growth by creating a surface concentration gradient that induces crystal growth.² We prepared liposomes of varying surface electrostatic composition by doping the lipid bilayers with

the cationic lipid 1,2-dioleoyl-3-trimethylammonium propane (DOTAP). These liposome formulations were incubated in both 20 mM and 40 mM zinc for 60 min, resembling the concentrations tested in our reported experimental procedures. The mixtures were then washed three times to remove unbound zinc, and the pellet digested in nitric acid. Results show Zn association to the liposome surface only in the case of negatively charged anionic liposomes (3.4–3.6 mM Zn²⁺ concentrations for 20 mM and 40 mM samples). On the contrary, in the case of positively charged liposomes the pellets showed only residual zinc association as per ICPMS quantification (~ 2 orders of magnitude less).

Figure S7: Investigation of ZIF growth around liposome and proteoliposome formulations. A) Zeta potential plot of pristine liposomes, encapsulated liposomes, and ZIF-L control. Data also includes measurement of proteoliposomes@ZIF and the respective control. B) Zeta potential values of pristine liposomes, encapsulated liposomes, ZIF-L control, proteoliposomes@ZIF, ZIF-L (Proteoliposome control). C) ICPMS Zn count plot for anionic (AnLp) and cationic (CatLp) incubated in varying concentrations of the metal precursor. D) Count per second and Zn concentrations (mM) for anionic (AnLp) and cationic (CatLp) liposomes exposed to Zn solutions.

This interaction is similar to the previously reported association between negatively charged proteins surface and zinc cations that is crucial for the successful biomimetic mineralization of ZIF-8 around proteins.^{3,4} We plan to use this information to investigate ZIF growth around cationic liposomes for the precise determination of liposome-based ZIF composites with different topologies in subsequent papers. Indeed, as shown by Carraro et al. this study required the investigation of 72 samples in triplicates to determine the influence of compositional variables on the crystalline phase of protein based ZIF composites.⁵

Kinetics of nucleation, particle growth, crystallization, and morphology of the particles were investigated *in situ* via small-angle and wide-angle X-ray scattering (SAXS/WAXS) at the Elettra

Synchrotron (Trieste – Italy). Results show that in the presence of liposomes, particle nucleation and growth is faster than the control samples (i.e. no liposomes). Indeed, the 20x16 Lp@ZIF composite shows a growth 5 times faster than that of the 20x16 ZIF-L control. It is noteworthy that, in the 0.1-120 s time interval for 20x16 Lp@ZIF and 0.1-50 s time interval for 40x16 Lp@ZIF, the material was amorphous. This shows the absence of crystalline materials during the particle nucleation and subsequent growth; a similar process was observed when composite particles were obtained by mixing proteins with the same MOF precursors.⁶ Then, we observed a faster crystallization for liposome@MOF particles than for the controls (e.g. 20x16 Lp@ZIF crystallization is 15 times faster than the pure MOF particles). By fitting the SAXS patterns, the presence of the liposome induced the formation of plate-like particles with a thickness of 30-50 nm. Conversely, in absence of liposomes, MOF particles with an average size bigger than 100 nm and with no sharp size distribution were observed. In summary, these *in situ* synchrotron investigations demonstrate that liposomes act as templating agents for the MOF growth and that liposome@MOF biocomposites are formed via the biomimetic mineralization process.^{3,7}

Details of the insitu experiments

To examine the role of liposome in the MOF formation, we investigated four different samples (20x16 Lp@ZIF, 20x16 ZIF-L, 40x16 Lp@ZIF and 40x16 ZIF-L) by using a stopped flow device for the rapid mixing of the reagents (<100 ms) and monitoring the reactions with a time-resolved SAXS/WAXS synchrotron set-up. The injection of the aqueous precursors solutions (Zn^{2+} , 2-methylimidazole, liposomes) into a micromixer triggered the acquisition system of rapid SAXS data collection with a time resolution of 100 ms. The volume ratio between the three precursor solutions was set to maintain the final concentrations used for the syntheses in batch.

Nucleation and growth kinetic: The Invariant \tilde{Q} (see SI experimental section for details), is related to the Porod invariant of the scattering curve. The increase of \tilde{Q} over time indicates the formation of particles/agglomerates within the investigated volume of the sample. A plateau in the time series of \tilde{Q} values, indicates stationary conditions. The time evolution of \tilde{Q} is reported in Figure S16 and the results are summarized in Table S5. The increase of \tilde{Q} related to the particle growth is detected 0.8 s and 0.6 s after the mixing of precursors for samples 20x16 Lp@ZIF and 40x16 Lp@ZIF, respectively. In the control samples, the particle growth is detected 4 s (20x16 ZIF-L) and 2.6 s (40x16 ZIF-L) after mixing the precursors. The plateau of \tilde{Q} related is detected 25 s and 5 s after the mixing of precursors for samples 20x16 Lp@ZIF and 40x16 Lp@ZIF, respectively. In the control samples, the plateau is reached 40 s (20x16 ZIF-L) and 25 s (40x16 ZIF-L) after mixing the MOF precursors. These data demonstrate that the presence of liposomes induces a quicker nucleation and faster particle growth when compared with the control samples (e.g. for 20x16 Lp@ZIF the crystallization plateau is reached is 5 times faster than 20x16 ZIF-L). Moreover, the kinetic of nucleation and growth is faster when a higher concentration of MOF precursors is used (e.g. for 40x16 Lp@ZIF the plateau is reached is 5 times faster than 20x16 Lp@ZIF).

Figure S16: Time evolution of Porod-Invariant \tilde{Q} (0.1–0.6 nm⁻¹ range) calculated from time-resolved SAXS synthesis of 20x16 LpZ and 20x16 Z (a) and of 40x16 LpZ and 40x16 Z (b). Selected SAXS patterns used for the calculation of \tilde{Q} are reported in **Figure S17**. The dashed lines are plotted to highlight the starting time of the \tilde{Q} increase.

Figure S17: Time evolution of SAXS patterns (background subtracted) from time-resolved SAXS synthesis of 20x16 Lp@ZIF (a), 20x16 ZIF-L (b), 40x16 Lp@ZIF (c) and 40x16 ZIF-L (d).

Table S5: Summary of the particle growth kinetic obtained from the analysis of the time evolution of Porod-Invariant \bar{Q} (0.1–0.6 nm⁻¹ range, **Figure S16**). Time zero is referred to the end of the precursors mixing.

	Particles growth start after:	Particles growth approach plateau after:
20x16 Lp@ZIF	0.8 s	25 s
20x16 ZIF-L	4 s	40 s
40x16 Lp@ZIF	0.6 s	5 s
40x16 ZIF-L	2.6 s	25 s

Crystallization kinetics: The crystallization kinetic was monitored following the integrated intensity of the (200) ZIF-L diffraction peak (5.25 nm^{-1} ; **Figure S18**). The results are summarized in Table S6. The crystallization of samples 40x16 Lp@ZIF and 40x16 ZIF-L were studied using the stopped flow set-up. Conversely, the crystallization of samples 20x16 Lp@ZIF and 20x16 ZIF-L were studied in vertically positioned capillary that was a manually filled with the a solution of premixed reagents (see experimental section for details, time from the mixing of the precursors to the first measurement = 120 seconds). This set-up was necessary because of the rapid flocculation and precipitation of the solid material out of the X-ray beam when using the horizontally mounted stop-flow capillary.

In the case of the 20x16 samples, in presence of liposomes, we detected the presence of the (200) ZIF-L diffraction peak (120 seconds after the mixing of the precursors; **Figure TU3a**) and we monitored its integrated area over time. Following this trend, 10 minutes after the mixing of the precursors a plateau of the integrated intensity of the (200) ZIF-L diffraction peak was observed. This indicates that crystallinity of the composite material has reached stationary conditions. Conversely, without the liposomes (control sample; **Figure S18 B**), the integrated intensity of the (200) ZIF-L diffraction peak showed to increase 29 minutes and plateau 35 minutes after the mixing of the precursors, respectively.

In the case of the 40x16 samples, in presence of liposomes, we observed the (200) ZIF-L diffraction peak 50 seconds after the mixing of the precursors (**Figure S18 C**). By monitoring the integrated area of this peak ad a function of time, we observed a plateau in 4 minutes. This indicates that changes in the crystallinity have stopped. Conversely, without the liposomes (control sample; **Figure S18 D**), the (200) ZIF-L diffraction peak was observed 60 seconds after the mixing of the precursors and a plateau was reached 12 minutes after the mixing of the MOF precursors.

These data demonstrate an absence of diffraction peaks in the early stage of particles growth (e.g. less than 120 s for 20x16 Lp@ZIF and less than 50 s for 40x16 Lp@ZIF), suggesting the initial formation of amorphous particles.⁶ Then, we observed that the presence of liposomes triggers a faster MOF crystallization when compared with the control samples.

Figure S18: Time evolution of the integrated intensity of (200) diffraction peak of ZIF-L (5.25 nm^{-1}) calculated from time-resolved SAXS synthesis of 20x16 Lp@ZIF (a), 20x16 ZIF-L (b), 40x16 Lp@ZIF (c) and 40x16 ZIF-L (d). In the insets, selected diffraction patterns highlighting the time-evolution of the (200) diffraction peak of ZIF-L (5.25 nm^{-1}) are reported. Time zero is referred to the end of the precursors mixing.

Table S6: Summary of the particle crystallization kinetic obtained from the analysis of the time evolution of the integrated intensity of (200) diffraction peak of ZIF-L (5.25 nm^{-1}); **Figure S18**. Time zero is referred to the end of the precursors mixing.

Sample	First detection of MOF 1 st diffraction peak (time after precursors mixing)	End of crystallization process (plateau of the integrated intensity of MOF 1 st diffraction peak; time after precursors mixing)
20x16 Lp@ZIF	<120 seconds	10 minutes
20x16 ZIF-L	29 minutes	35 minutes
40x16 Lp@ZIF	50 seconds	4 minutes
40x16 ZIF-L	60 seconds	12 minutes

Morphology of the particles: we modeled the SAXS patterns 60 seconds after the mixing of the reagents (e.g. plateau of \tilde{Q} ; Figure TU4) to investigate the morphology of the formed particles. The fits revealed that the liposomes-containing samples (20x16 Lp@ZIF and 40x16 Lp@ZIF) possess a plate-like structure (thickness 30-50 nm; lateral size > 100 nm; see experimental in SI section for details) that is preserved in the final material (see the morphology in the SEM images of samples prepared with a 24h of synthesis). Conversely, the control samples 20x16 ZIF-L and 40x16 ZIF-L can be fitted with a classical Porod power law with an exponent close to the theoretical value of 4 (3.7 for 20x16 Z and 3.9 for 40x16 Z), suggesting that, during our SAXS experiments, the particles possess an average size bigger than 100 nm with no sharp size distribution. As the presence of liposome influences the morphology of the crystals, as previously observed for other biomimetic mineralization agents,³ these data further support that the formation of 20x16 Lp@ZIF and 40x16 Lp@ZIF composites biomimetic mineralization mechanism.

Figure S19: SAXS patterns (background subtracted and averaged) and fitted data 120 seconds after mixing the precursors of 20x16 Lp@ZIF (a), 20x16 ZIF-L (b), 40x16 Lp@ZIF (c) and 40x16 ZIF-L (d). In c and d, the theoretical Porod power law $I(q) \propto q^{-4}$ is plotted for comparison.

Further, different metal to-ligand ratios were tested for biomolecular nucleation of ZIF in liposomes. The new conditions included the following: 20mM zinc-640 mM HMIM (1:32), 40mM zinc-1240 mM HMIM (1:32), and 20mM zinc-1240 mM HMIM (1:64). We found that increasing the metal-to ligand ratios results in formation of sodalite ZIF-8. The formation of sodalite ZIF-8 using high ligand:metal molar ratio (e.g. >32) is in line with previously reported syntheses of pure ZIF-8⁹ and of ZIF-8 biocomposites.^{2,5,10,11} This was confirmed by TEM and PXRD. Further, we were happy to find that in such conditions the liposomes could still be recovered as shown by TEM and DLS, although the removal of the shell was slightly more difficult. It is noteworthy that the interaction between the biomacromolecule and ZIFs depends on several factors including: The nature of the biomacromolecule (e.g. carbohydrates),¹¹ changes in the surface chemistry of the biomacromolecule,⁴ and proteins⁵ could lead to the formation of different ZIFs using the same

precursors concentration and metal:ligand molar ratio.⁴ Therefore, a dedicated study would be necessary to fully explore the nature of the liposome-ZIF interaction.

Figure S5: Crystal characterization of 20x32, 40x32, and 20x64 Lp@ZIF-8. A) SEM micrograph of 20x32 Lp@ZIF-8 and B) of 20x32 pristine ZIF-8. C) SEM micrograph of 40x32 Lp@ZIF-8 and D) of 40x32 pristine ZIF-8. E). SEM micrograph of 20x64 Lp@ZIF-8 and F) of 20x64 pristine ZIF-8. G) PXRD spectra collected from 20x32 Lp@ZIF-8, 40x32 Lp@ZIF-8, 20x64 Lp@ZIF-8, and corresponding controls.

Figure S6: Liposome recovery of 20x32, 40x32, and 20x64 Lp@ZIF-8. A) TEM micrograph of liposomes recovered after exfoliation of 20x32 Lp@ZIF-8 (Scale bar = 100 nm), B) of liposomes recovered after

exfoliation of 40x32 Lp@ZIF-8 (Scale bar = 50 nm), and C) of liposomes recovered after exfoliation of 20x64 Lp@ZIF-8 (Scale bar = 100 nm). E). DLS exfoliated 20x32 Lp@ZIF-8, 40x32 Lp@ZIF-8, and 20x64 Lp@ZIF-8

According to these new experimental investigation and results we have expanded the manuscript text and SI file to report these new finding which nicely complement our initial submission.

We are truly grateful to the reviewer for fexpanding our study towards a more detailed investigation on the molecular details of the ZIF biomineralization process which we feel further expanded the impact of the work.

- Related to the points above, the crystalline, ZIF-8 exoskeleton is not a phase-pure MOF material, which is what has been observed in other protein encapsulation studies before. The authors should provide a more detailed analysis and explanation for the PXRD spectrum in Figure 1B. Other related questions: If the new reflections peaks are formed during encapsulation in the case of the 20 x 16 ZIF-8, why does the control have the same new peaks as 20 x 16 LP @ ZIF-8? In the case of 40 x 16 ZIF-8, if it matches poorly with the known polymorphs of ZIF-8, what are the any other possible structural models?

Importantly, why is the PXRD of 40 x 16 ZIF-8 in Figure 2F is so different from 40 x 16 ZIF-8 in Figure 1B? On that note, the blue and red traces in Figure 2F look “too identical”.

We are grateful for the observation. The reviewer inspired us to revisit the synthesis and activation procedure to suss out some of the points they have been raised. In brief: we found our activation procedure needed to be refined. Data collection for the PXRD, porosity, and TGA were conducted on “activated” samples that had been washed with methanol, which is standard proceeedure in MOF activation. Our original activation procedure was inducing a phase transition in some samples because the length of soaking had not been standardized. We have since made changes to our activation procedure and this procedure is now more detailed in the supporting information. From our recently obtained results, we conclude the following: Our reaction affords the formation of a previously reported zeolite phase known as “leaf”, commonly referred as ZIF-L. This is true for every composite reported in our original manuscript (liposomes, transmembrane proteins, and proteoliposomes). In depth analysis of the obtained phase nicely matches with XRD results reported elsewhere.^{2,3} After revision of our synthetic parameters we realized that, prolonged exposure to methanol (during washing steps and/or particle activation for structural analysis) can result in phase transition. This phenomenon has previously been reported by others.⁴

Figure 1. Characterization of artificial lipid bilayers embedded in ZIF. PXRD spectra of ZIF liposome complexes (Lp@ZIF) and ZIF controls.

Figure 3. IroT@ZIF and CopA@ZIF characterization. PXRD spectra of ZIF liposome complexes (Lp@ZIF) and ZIF controls.

Figure S11. Characterization of IroT/CopA-PL@ZIF. PXRD spectra of protein-PL@ZIF complexes (CopA-PL@ZIF and IroTPL@ZIF) and ZIF-L controls.

- The TGA of the 40 x 16 LP@ZIF sample shows a gradual ~30% mass loss starting around 120 C, not 200 C (as stated in the text). If this is attributed to the decomposition of lipids, it is not clear to me as to why it is so different than what is observed in the case of the 20 x 16 LP@ZIF (especially given that the 40 x 16 ZIF-8 control is thermally more stable than the 20 x 16 ZIF-8 control).

This is a great observation as it was a consequence of the phase impure material induced from our prior methanol wash. We are extremely grateful to the reviewer for pointing this out! We reran thermogravimetric analysis of both 40x16 and 20x16 Lp@ZIF and controls using our

refined activation process. Results now show similar behavior to that of previously reported ZIF-L, as expected given the PXRD results shown above.

Figure S2. Characterization of ZIF-L and ZIF-L liposome composites. Thermogravimetric analysis of 40x16 Lp@ZIF, 40x16 ZIF-L, 20x16 ZIF-L, and 20x16 Lp@ZIF.

- I don't see a correlation between porosity (or for that matter, crystallinity) and protein stabilization. So, what is then the primary determinant of protein stabilization and if there is no correlation, would it be simply ok to have an amorphous coat on the surface? Further regarding porosity: it would be interesting to see if CopA and Iro were catalytically active while encapsulated.

As the reviewer noted, there is no relationship between porosity and stabilization per se however, literature has shown a correlation between protein stability and ZIF phase. (we have briefly discussed this in this response and now in the manuscript—the phase dependence on stability is an active area of research. Amorphous ZIF coordination materials do protect proteins; however, these shells are kinetically very labile and thermodynamically unstable.¹⁰ More work will be done in this space in subsequent papers. Catalytic activity while encapsulated is an exciting idea! However, ZIF-L is essentially impermeable to anything but CO₂,² though other phases may offer larger pores, which can expand our process even more.

Sample ID	BET surface (m ² /g)
20x16 ZIF-L	73
20X16 Lp@ZIF-L	38
40x16 ZIF-L	385
40X16 Lp@ZIF-L	288

Figure S2. Characterization of ZIF-L and ZIF-L liposome composites. Nitrogen isotherms of 40x16 Lp@ZIF, 40x16 ZIF-L, 20x16 ZIF-L, and 20x16 Lp@ZIF.

-It would be beneficial to directly show protein/liposome encapsulation via fluorescence microscopy.

As recommended, Cy5 was encapsulated within the lumen of the liposomes and the resulting formulation used for biomolecular nucleation in ZIF-8. Supernatants were collected after first washing and used for determination of encapsulation efficiency. Washed crystals were then mounted in a microscope slide and observed using a confocal microscope. The crystals, which are 150-200nm and not much larger than the liposomes themselves, are fully fluorescent.

Figure S1. Encapsulation efficiency determination of liposomes-embedded in ZIF-L. A) Fluorescence emission spectra of Cy5-loaded liposome versus supernatants collected for 20x16 Lp@ZIF, 40x16 Lp@ZIF. Controls include supernatants collected for both 20x16 and 40x16 ZIF-L. B) Confocal microscope caption of 20x16 and 40x16 ZIF-L. C) Confocal microscope caption of 20x16 Lp@ZIF. D) Confocal microscope caption of 40x16 Lp@ZIF. Pristine ZIF-L shows no intrinsic fluorescence when imaged under the Cy5 channel. On the other hand, both 20x16 Lp@ZIF and 40x16 Lp@ZIF prepared with Cy5-loaded liposomes have fluorescence in the cyanine channel (670 nm).

-“aiding and abetting” is an odd term to use. It is perhaps more appropriate for a courtroom than a scientific article.

Main text now includes a more contemporary term.

- Refs 37, 41 need to be corrected.

References have been readdressed and corrected according to the reviewer recommendation.

Reviewer #3 (Remarks to the Author):

This manuscript describes an excellent piece of research that truly advances the field of MOF biomimetic mineralisation, and, more broadly, shows how MOF-based composites can be used to stabilise proteoliposomes. The value of this result is evidenced by a series of experiments that show the

composites can protect relatively fragile membrane associated proteins from stressors that typically lead to their degradation. Hitherto, biomimetic mineralisation has focused on encapsulating proteins that are far more robust and thus, this work significantly broadens the scope of this area. I think the work will be of particular interest to both materials scientists and biochemists.

I enthusiastically recommend this paper for publication in Nature Commun. as in my opinion it clearly meets the required novelty and scholarly approach required for the journal. Nevertheless, prior to acceptance there are some minor comments that I think the authors should address,

1. The authors mention, rightly, that the formation of ZIF-based biocomposites is influenced by the surface charge of the biomacromolecule (in this case the liposome). Ref 37 is incomplete, further, also the authors may want to include the following paper as it shows that the notion applies to other biomolecules Mater. Horiz., 2019, 6, 969-977. In addition, did the authors measure zeta potentials? Not necessary if they don't have the data but would be useful to consider in future work

We appreciate the recommendation by the reviewer. We now have determined zeta potentials of both liposomes, proteoliposomes, and their respective controls. Information is now included in the manuscript.

Figure S7: Investigation of ZIF growth around liposome and proteoliposome formulations. A) Zeta potential plot of pristine liposomes, encapsulated liposomes, and ZIF-L control. Data also includes measurement of proteoliposomes@ZIF and the respective control. B) Zeta potential values of pristine liposomes, encapsulated liposomes, ZIF-L control, proteoliposomes@ZIF, ZIF-L (Proteoliposome control).

2. The isotherms shown in figure S1C are described as Type II in the manuscript. To me these are all

clearly Type I. Type II isotherms have a characteristically increasing uptake which is representative of a high external:internal surface area ratio. Adsorbate condensation at saturation pressures is common where there are macro/mesoporous gaps between crystals or if large crystals are cracked. In light of this perhaps the condensation might result from a heterogeneous coating? i.e phase boundaries (ZIF-8/ZIF-C) or microcrystalline composites rather than a single crystal? The authors may wish to change their interpretation of the data for this section.

The reviewer is correct, of course, the isotherms are indeed Type I. The incorrect phrase has been removed.

REFERENCES

1. Low, Z.-X et al. Crystal transformation in zeolitic-imidazolate framework. *Cryst. Growth Des.* **14**, 6589-6598 (2014).
2. Li, S. et al. Investigation of controlled growth of metal-organic frameworks on anisotropic virus particles. *ACS Appl. Mater. Interfaces* **10**, 18161-18169 (2018).
3. Liang, K. et al. Biomimetic mineralization of metal-organic frameworks as protective coatings for biomacromolecules. *Nat. Commun.* **6**, 7240 (2015).
4. Maddigan, N.K. et al. Protein surface functionalisation as a general strategy for facilitating biomimetic mineralisation of ZIF-8. *Chem. Sci.* **9**, 4217-4223 (2018).
5. Carraro, F. et al. Phase dependent encapsulation and release profile of ZIF-based biocomposites. *Chemical Science* **11**, 3397-3404 (2020).
6. Carraro, F. et al. Continuous-flow synthesis of ZIF-8 biocomposites with tunable particle size. *Angew. Chem. Int. Ed.* **59**, 8123-8127 (2020).
7. Liang, W. et al. Enzyme Encapsulation in a Porous Hydrogen-Bonded Organic Framework. *Journal of the American Chemical Society* **141**, 14298-14305 (2019).
8. Jian, M. et al. Water-based synthesis of zeolitic imidazolate framework-8 with high morphology level at room temperature. *RSC Advances* **5**, 48433-48441 (2015).
10. Liang, W. et al. Control of structure topology and spatial distribution of biomacromolecules in protein@ZIF-8 biocomposites. *Chem. Mater.* **30**, 1069-1077 (2018).
11. Astria, E. et al. Carbohydrates@MOFs. *Mater. Horiz.* **6**, 969-977 (2019).

REVIEWERS' COMMENTS

Reviewer #1 (Remarks to the Author):

The authors have satisfactorily addressed my previous concerns. The manuscript can be published in its present form.

Reviewer #2 (Remarks to the Author):

I thank the authors for their efforts to address my comments (and those of the other reviewers). I believe that they did a good job with their revision. There are still some questions/issues remaining, but as the authors state, these can be dealt with in future studies. In sum, this work is a valuable and exciting addition to the literature on enzyme encapsulation.

Response to Reviewers. Our response is in BLUE.

REVIEWERS' COMMENTS

Reviewer #1 (Remarks to the Author):

The authors have satisfactorily addressed my previous concerns. The manuscript can be published in its present form.

Thanks!

Reviewer #2 (Remarks to the Author):

I thank the authors for their efforts to address my comments (and those of the other reviewers). I believe that they did a good job with their revision. There are still some questions/issues remaining, but as the authors state, these can be dealt with in future studies.

In sum, this work is a valuable and exciting addition to the literature on enzyme encapsulation.

Thanks!